

**The influence of near surface sediment hydrothermalism on the TEX86 tetraether**
**lipid-based proxy and a new correction for ocean bottom lipid overprinting**

Jeremy N. Bentley [a], Gregory T. Ventura [a], Clifford C. Walters [b], Stefan M. Sievert [c],
Jeffrey S. Seewald [c]
[a] *Department of Geology, Saint Mary's University, Halifax, Nova Scotia B3H 3C3,*
*Canada.*
[b] *Bureau of Economic Geology, University of Texas at Austin, USA.*
[c] *Woods Hole Oceanographic Institution, Woods Hole, USA.*
* *Corresponding author:* *Todd.ventura@smu.ca*
**Key Points**
• High *i*GDGTs turnover in shallow sediments is shown to be non-selective and
does not impact TEX86 paleoclimate ratios.
• The proxy nonetheless can be overprinted by addition of sediment sourced lipids
when geothermal temperatures rise above ~60–70 °C.
• A universally applicable, diagenetic correction model is presented to remove
overprinting artifacts in the TEX86 proxy.

**Abstract**

The diversity and relative abundances of tetraether lipids produced by Thaumarchaeota in
soils and sediments increasingly is used to assess environmental change. For instance, the
TetraEther indeX of 86 carbon atoms (TEX86), based on archaeal isoprenoidal glycerol
dialkyl glycerol tetraether (*i*GDGT) lipids, is frequently applied to reconstruct past sea-
surface temperatures (SST). Yet, it is unknown how the ratio fully responds to
environmental and or geochemical variations and if the produced signals are the adaptive
response by Thaumarchaeota to climate driven temperature changes in the upper water
column. We present the results of a four push-core transect study of surface sediments
collected along an environmental gradient at the Cathedral Hill hydrothermal vent system
in Guaymas Basin, Gulf of California. The transect crosses a region where advecting



hydrothermal fluids reach 155 °C within the upper 21cm below the seafloor (cmbsf) close
to the vent center to near ambient conditions at the vent periphery. The recovered $i$GDGTs
closest to the vent center experienced high rates of turnover with up to 94% of lipid pool
being lost within the upper 21 cmbsf. Here, we show that turnover is non-selective across
$TEX_{86}$ GDGT lipid classes and does not independently affect the ratio. However, as evident
by $TEX_{86}$ ratios being highly correlated to the Cathedral Hill vent sediment porewater
temperatures ($R^2 = 0.84$), the ratio can be strongly impacted by the combination of severe
lipid loss when it is coupled to the addition of $in\ situ$ $i$GDGT production from archaeal
communities living in the vent sediments. The resulting signal overprint produces absolute
temperature offsets of up to 4 °C based on the $TEX_{86}^H$ -calibration relative to modern climate
records of the region. The overprint is also striking given the flux of GDGTs from the upper
water column that is estimated to represent ~93% of the combined intact polar lipid (IPL)
and core GDGT lipid pool initially deposited on the seafloor. A model to correct the
overprint signal using IPLs is therefore presented that can similarly be applied to all near-
surface marine sediment systems where calibration models or climate reconstructions are
made based on the $TEX_{86}$ measure.

**1. Introduction**
Archaeal and bacterial tetraether cellular membrane lipids mark common and structurally
diverse compounds that are frequently used to track the presence of living and dead
microorganisms in the geosphere (e.g. Schouten et al., 2002, 2004; Hopmans et al., 2004;
Weijers et al., 2007; Lipp et al., 2008). The proportional abundances of these lipids forms
various prominent proxies for assessing environmental change through time. For example,
$TEX_{86}$ (TetraEther indeX with 86 carbon atoms (Schouten et al. (2002) is the most widely
used archaeal lipid-based paleotemperature proxy for marine environments (Table 1; Eq.
1). This proxy measures variations in the number of cyclopentyl rings within the
hydrocarbon skeleton of a select range of archaeal core lipid (CL) classes (Supplementary
Figure A-1) following the initial assumption that cyclization of the biphytanyl moiety is an
organismal response to changing sea surface temperatures (SSTs). The proxy is therefore
used in paleo-oceanographic studies in many different regions around the world (Huguet et
al., 2006; Kim et al., 2008; McClymont et al., 2012) with $TEX_{86}$ values typically ranging
from 0.2–0.9 in both marine and lake sediment (Sinninghe Damsté et al., 2009; Powers et
al., 2010; Zhang et al., 2016 Morrissey et al., 2018; Yao et al., 2019; Kumar et al., 2019).
The utility of $TEX_{86}$ rests on the premise that $i$GDGTs found in ocean bottom sediments
are almost exclusively produced by marine planktonic Thaumarchaeota that inhabit the
epipelagic zone (Wakeham et al., 2003; Tierney, 2014).$TEX_{86}$-based lipids are therefore
required to be efficiently and continually transported from the upper water column to the
underlying ocean floor sediments to produce a chemostratigraphic record of microbial
response to changing SST conditions (Wuchter et al., 2005).

Since its introduction, the reliability of $TEX_{86}$ to accurately track paleoclimate variations
has been questioned. For example, over the past decade, considerable effort has been made
to reconstruct the early Paleogene greenhouse climate with a variety of paleoclimate
proxies (Hollis et al., 2012). However, $TEX_{86}$ appears to significantly over-estimate
reconstructed SSTs relative to other proxies such as Mg/Ca, clumped isotopic compositions



of foraminiferal calcite, as well as various climate models based on partial pressure of
carbon dioxide ($p$CO$_2$) predictions (Lunt et al., 2012; Naafs et al., 2018). The apparent high
SST reconstructions have been attributed to proxy complications including ocean
subsurface sediments origin of lipids (Ho and Laepple, 2016). For late Neogene climate
reconstructions, the proxy has been shown to underestimate warming trends relative to $U_{37}^{k\prime}$-
derived temperatures (Lawrence et al., 2020). In this regard, the debate largely centers on
a lack of understanding of how the proxy's associated lipids precisely change in relation to
their environment and if these changes are regulated by internal adaptations within the
archaeon or by community succession (Elling et al., 2015; Qin et al., 2015).
Additionally, most Thaumarchaeota are found below the photic and epipelagic zone and
should therefore not produce a direct response to changing SSTs. Studies from the Pacific
Ocean have shown that peak archaeal abundances occur at 100–350 m depth (Karner et al.,
2001; Pearson et al 2013). To address the impact of depth habitat, Schouten *et al.* (2013)
further proposed a calibration based on suspended particulate matter and *in situ* water
temperature from the upper 100 m of the global ocean. In this regard, if these deeper
sourced lipids are deposited on the seafloor than the sedimentary GDGT used to generate
sea surface temperatures are mixed with significant contributions from much colder waters
potentially impacting the reconstructed values providing much lower SSTs. As TEX$_{86}$ may
become disproportionality impacted by the collection of mixed source inputs; the location
of lipid loading from the water column to the ocean floor sediments seems to be an factor
as strong positive relationship between water depth and differences in TEX$_{86}^{H}$ values are
observed in both surface sediments and suspended particulate organic matter from the
Mediterranean Sea (Kim et al., 2015). The differences appear to be driven by increase
relative abundances of TEX$_{86}$ lipids GDGT-2 and the isomers of crenarchaeol (Lui et al.,
2018; Damsté et al., 2018) coupled to decreasing abundances of GDGT-1 and GDGT-3
with increasing water depth. The systematic change results in a higher reconstructed SST
bias for deep-water surface sediments. Therefore, such sourcing effects have further
resulted in speculation that the TEX$_{86}$ ratio of open ocean sediments may actually reflect
deeper water column and subsurface rather than SSTs (Huguet et al., 2007; Lopes dos
Santos *et al.*, 2010; Kim *et al.*, 2012a,b; Ho & Laepple, 2016; Hurley et al., 2016). To
compensate for this, both TEX$_{86}^{H}$ and TEX$_{86}^{L}$ have been re-calibrated against subsurface (0–
900 m water depth) temperatures (Kim *et al.*, 2012a,b; Ho & Laepple, 2016).
The GDGT relative abundances recorded in a TEX$_{86}$ measurements may constitute a multi-
variable system, having both a component of lipids contributed to the "pool" via *in situ*
sources and by depositional processes. TEX$_{86}$-based SST estimates have been observed to
substantially deviate from other temperature proxies (e.g., Huguet et al., 2006; Liu et al.,
2009; Rommerskirchen et al., 2011; Hollis et al., 2012; Seki et al., 2012) implying these
values can be a response to seasonal biases, non-thermal influences, or other ecological
signals. Non-thermal influence result from lipid abundances being brought to marine
sediments from non-planktonic Thaumarchaeota origins such as from the deep water or
within marine sediments (Liu et al., 2011; Kerou et al. 2020). There are likely other driving
forces other than temperature that impact the archaeal GDGT production and relative
abundances. Some examples of these drivers include organismal selectivity to specific



growth phases and growth rates (Elling et al., 2014; Hurley et al., 2016); ammonia
oxidation rates (Hurley et al., 2016); and redox conditions (Qin et al., 2015).

By artificially hydrolyzing the headgroups of marine archaeal IPLs harvested from a
sediment trap, Lipp and Hinrichs (2009) demonstrated that the production of GDGTs by
ocean floor sediment microbial communities may impact $TEX_{86}$ values. Similarly, Elling
et al. (2015) confirmed $TEX_{86}$ values can represent a mixed GDGT signal from both active
microbial production in shallow sediments and fossil lipids sourced from the water column.
These authors further demonstrated that $TEX_{86}$ values from cultures can diverge from the
global calibration that forms the basis for most climate reconstructions suggesting that the
sedimentary community compositions may exert some controls on the $TEX_{86}$ signal.
Besseling et al. (2019) further extended these concerns, suggesting $TEX_{86}$ reflects
subsurface temperatures rather than SSTs as the input of GDGTs in marine settings are not
exclusive to Thaumarchaeota, because a majority of marine group I (MGI) Archaea also
reside in subsurface waters or marine sediments. Collectively, these observations indicate
a sub-pelagic zone where microorganisms may mix with the GDGTs from the surface, thus
providing mixed signals and inaccurate $TEX_{86}$ values from mixed sources.  However, other
authors have found that $TEX_{86}$ ratios are not impacted by benthic Archaea due to the low
relative turnover rates for the lipids in marine sediments (Lengger et al., 2012, 2014; Omuh
et al., 2020). Omuh et al. (2020) found little effect to the $TEX_{86}$ paleoclimate ratio when
examining surface sediments near hydrothermal vent sites on the Southeast Indian Ridge
in the southern Indian Ocean. Lengger et al. (2012, 2014) reported no significant deviation
between the $TEX_{86}$ values in sediment cores collected near the oxygen minimum zone from
that of the overlying water column in the Arabian Sea with near linear degradation rates of
both IPLs and CLs.

While not an ideal location to create SST reconstructions, hydrothermal vents of
sedimented ocean basins do represent an anomalous end-member to the vast expanse of
ambient ocean floor sediment where paleoclimate reconstructions are commonly produced.
The Guaymas Basin, Gulf of California (Figure 1) is one such site. The basin experiences
elevated sedimentation rates ranging between 0.4–0.2 cm/yr. (Curray et al., 1979; Gieskes
et al., 1988) due in part to the high productivity of the upper water column. The ocean floor
hydrothermally impacted surface sediments are also a location of active and diverse
microbial communities with vents that are often covered by Beggiatoa dominated microbial
mats (Teske et al., 2016). These sites should in principle enable a high-resolution archaeal
lipid-based paleoclimate record that provides optimal conditions for studying potential
subsurface lipid overprinting or interferences to common archaeal lipid-based
environmental proxies. For this study, we examined near-surface ocean floor sediments
from the Cathedral Hill hydrothermal vent complex (Figure 1) in the Guaymas Basin to
determine if sea surface paleoclimate proxy signals can be impacted by the presence of
subsurface archaeal populations. The distribution of GDGTs and their corresponding
environmental proxy signals were measured within the sediments along a transect at the
Cathedral Hill hydrothermal vent system. In this regard, this site offers the unique
opportunity to evaluate the response of $TEX_{86}$ and other tetraether-lipid proxies within a
microbially diverse sedimentary environment that is exposed to high temperature vent
fluids.



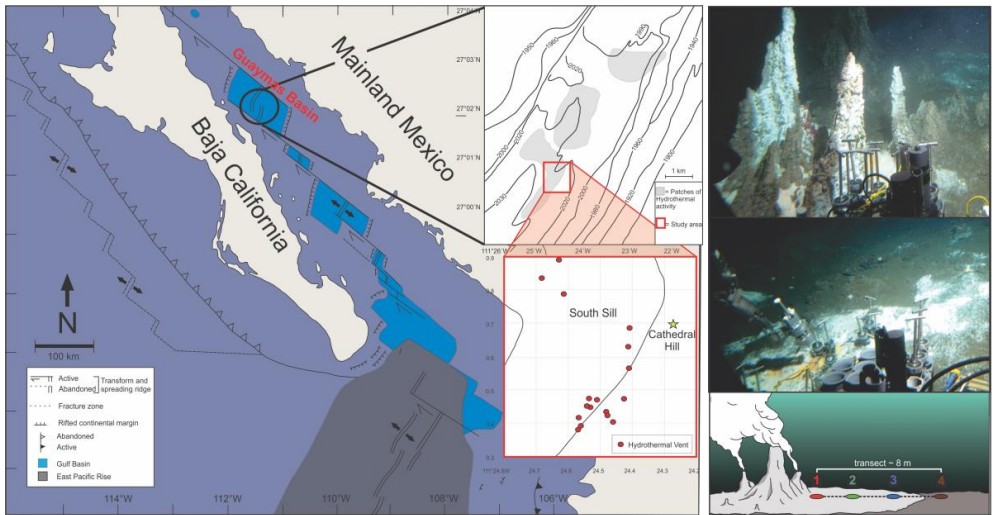


**FIGURE 1** A) Location map of Guaymas Basin and the Southern Sill (red outlined box) in the Gulf of California. Cathedral Hill is marked with a yellow star. B) Photo of Cathedral Hill taken via *Alvin*. C) Schematic of the push core transect with a color coding that is consistent for all plots throughout this paper. Maps modified from Teske et al. (2016) and Dazell et al. (2021).


**2. Material and methods**
**2.1. Study location and sampling**
Four sediment push cores were collected using HOV *Alvin* (Dive 4462; 10/22/08) at the
Cathedral Hill hydrothermal vent site, located at a water depth of 1996 m in the Southern
Trough of Guaymas Basin, Gulf of California (27°0.629' N, 111°24.265' W) (Figure 1).
The push cores, labeled 1 to 4, were taken along a transect with ~ 2 m spacing extending
outwards from microbial mat-covered sediments near the sulfide chimney complex to just
outside of the microbial mat area in ambient seafloor sediment. Thermal-probe
measurements were taken next to each core (Table 1). Once the push cores were brought
to the surface, they were subsampled into 2–3 cm-thick intervals, transferred to combusted
glass vials and immediately stored at -40 °C (onboard the ship) before being shipped under
dry ice to the laboratory and later freeze-dried and stored at -80 °C until being later
processed.





**Table 1.** Sediment geochemical and lipid proxy data.

| Core[*a] | Depth interval (cmbsf) | Alvin dive # and core ID | Description/lithology[*b] | Pore water temperature (°C) | Interpolated Pore water temperature (°C) | Sediment weight (g) | TLE μg/g sediment |
|---|---|---|---|---|---|---|---|
| 1 | 0-2 | GB4462-5 | Black mud with microbial mat filaments | 19 | 19 | 2.40 | 11552.3 |
| 1 | 2-4 | GB4462-5 | Brownish-green diatomaceous mud | - | 67 | 2.10 | 7648.2 |
| 1 | 4-6 | GB4462-5 | Brownish-green diatomaceous mud | 85 | 85 | 2.04 | 9266.0 |
| 1 | 6-8 | GB4462-5 | Brownish-green diatomaceous mud | - | 105 | 2.83 | 2088.3 |
| 1 | 8-10 | GB4462-5 | Brownish-green diatomaceous mud | - | 117 | 2.48 | 4378.1 |
| 1 | 10-12 | GB4462-5 | Grayish-green mud | 121, 124 | 125 | 2.52 | 1972.2 |
| 1 | 12-15 | GB4462-5 | Brownish-green consolidated mud with clay shards | - | 135 | 2.62 | 1992.4 |
| 1 | 15-18 | GB4462-5 | Brownish-green consolidated clay | 142 | 145 | 3.01 | 1691.0 |
| 1 | 18-21 | GB4462-5 | Brownish-green consolidated clay | 153 | 153 | 2.94 | 1722.0 |
| 2 | 0-2 | GB4462-6 | Black mud with microbial mat filaments | 9, 13 | 11 | 2.12 | 8476.2 |
| 2 | 2-4 | GB4462-6 | Black mud with microbial mat filaments | - | 22 | 2.30 | 8653.5 |
| 2 | 4-6 | GB4462-6 | Brownish-green diatomaceous mud | 20 | 20 | 3.30 | 2509.2 |
| 2 | 6-8 | GB4462-6 | Brownish-green diatomaceous mud | - | 47 | 2.84 | 3383.8 |
| 2 | 8-10 | GB4462-6 | Brownish-green diatomaceous mud | - | 60 | 3.34 | 1480.5 |
| 2 | 10-12 | GB4462-6 | Brownish-green diatomaceous mud | 69, 77 | 73 | 2.39 | 4185.9 |
| 2 | 12-15 | GB4462-6 | Brownish-green diatomaceous mud | - | 87 | 3.50 | 1694.3 |
| 2 | 15-18 | GB4462-6 | Brownish-green diatomaceous mud | 118 | 105 | 3.50 | 2011.6 |
| 2 | 18-21 | GB4462-6 | Brownish-green diatomaceous mud | 109 | 125 | 3.48 | 1382.2 |
| 3 | 0-2 | GB4462-3 | Black mud with microbial mat filaments | 3.2 | 3.2 | 2.81 | 7313.2 |
| 3 | 2-4 | GB4462-3 | Brownish-green diatomaceous mud | - | 8 | 2.88 | 3909.7 |
| 3 | 4-6 | GB4462-3 | Brownish-green diatomaceous mud | 15 | 15 | 2.45 | 2864.8 |
| 3 | 6-8 | GB4462-3 | Brownish-green diatomaceous mud | - | 26 | 2.80 | 5003.6 |
| 3 | 8-10 | GB4462-3 | Brownish-green diatomaceous mud | 34 | 34 | 2.80 | 2018.0 |
| 3 | 10-12 | GB4462-3 | Brownish-green diatomaceous mud | - | 43 | 3.15 | 1863.5 |
| 3 | 12-15 | GB4462-3 | Brownish-green diatomaceous mud | - | 54 | 3.15 | 1777.8 |
| 3 | 15-18 | GB4462-3 | Brownish-green diatomaceous mud | 61 | 66 | 2.45 | 1428.6 |
| 3 | 18-21 | GB4462-3 | Brownish-green diatomaceous mud | 83 | 80 | 2.80 | 1982.0 |
| 4 | 0-2 | GB4462-8 | Black mud | 0 | 0 | 2.80 | 3440.4 |
| 4 | 2-4 | GB4462-8 | Brownish-green diatomaceous mud | 1.5 | 8 | 2.80 | 3166.1 |
| 4 | 4-6 | GB4462-8 | Brownish-green diatomaceous mud | 16 | 16 | 2.55 | 4000.0 |
| 4 | 6-8 | GB4462-8 | Brownish-green diatomaceous mud | - | 18 | 2.80 | 4185.5 |
| 4 | 8-10 | GB4462-8 | Brownish-green diatomaceous mud | - | 21 | 3.33 | 4755.3 |
| 4 | 10-12 | GB4462-8 | Brownish-green diatomaceous mud | - | 23 | 2.44 | 4843.6 |
| 4 | 12-15 | GB4462-8 | Brownish-green diatomaceous mud | - | 25 | 0.32 | 5741.9 |
| 4 | 15-18 | GB4462-8 | Sample lost during collection | - | - | - | - |
| 4 | 18-21 | GB4462-8 | Sample lost during collection | 29 | - | - | - |

**Table 1.** Sediment geochemical and lipid proxy data (continued).

| Core[a] | Depth interval (cmbsf) | Alvin dive # and core ID | TEX$_{86}$ Core GDGT[c] | TEX$_{86}^{H}$ Core GDGT[d] | TEX$_{86}^{H}$ Reconstructed SSTs (Kim et al., 2010)[e] | RI[f] | MI[g] | TEX$_{86}$ 1G-GDGT[c] | TEX$_{86}$ Core GDGT[c] |
|---|---|---|---|---|---|---|---|---|---|
| 1 | 0-2 | GB4462-5 | 0.56 | -0.25 | 21.2 | 2.44 | 0.34 | 0.58 | 0.56 |
| 1 | 2-4 | GB4462-5 | 0.58 | -0.23 | 22.6 | 2.45 | 0.38 | 0.58 | 0.58 |
| 1 | 4-6 | GB4462-5 | 0.58 | -0.24 | 22.3 | 2.48 | 0.36 | 0.55 | 0.58 |
| 1 | 6-8 | GB4462-5 | 0.58 | -0.24 | 22.2 | 2.55 | 0.35 | 0.57 | 0.58 |
| 1 | 8-10 | GB4462-5 | 0.59 | -0.23 | 22.9 | 2.60 | 0.34 | 0.72 | 0.59 |
| 1 | 10-12 | GB4462-5 | 0.57 | -0.25 | 21.8 | 2.63 | 0.31 | 0.70 | 0.57 |
| 1 | 12-15 | GB4462-5 | 0.61 | -0.22 | 23.8 | 2.65 | 0.37 | 0.69 | 0.61 |
| 1 | 15-18 | GB4462-5 | 0.61 | -0.22 | 23.9 | 2.66 | 0.36 | - | 0.61 |
| 1 | 18-21 | GB4462-5 | 0.63 | -0.20 | 24.9 | 2.66 | 0.38 | - | 0.63 |
| 2 | 0-2 | GB4462-6 | 0.55 | -0.26 | 20.6 | 2.524 | 0.32 | 0.46 | 0.55 |
| 2 | 2-4 | GB4462-6 | 0.54 | -0.27 | 20.4 | 2.524 | 0.32 | 0.58 | 0.54 |
| 2 | 4-6 | GB4462-6 | 0.54 | -0.27 | 20.4 | 2.525 | 0.33 | 0.60 | 0.54 |
| 2 | 6-8 | GB4462-6 | 0.56 | -0.25 | 21.5 | 2.677 | 0.29 | 0.71 | 0.56 |
| 2 | 8-10 | GB4462-6 | 0.58 | -0.25 | 21.7 | 2.695 | 0.29 | 0.70 | 0.58 |
| 2 | 10-12 | GB4462-6 | 0.57 | -0.24 | 21.9 | 2.712 | 0.28 | 0.68 | 0.57 |
| 2 | 12-15 | GB4462-6 | 0.57 | -0.24 | 21.9 | 2.734 | 0.28 | 0.73 | 0.57 |
| 2 | 15-18 | GB4462-6 | 0.58 | -0.23 | 22.6 | 2.680 | 0.31 | - | 0.58 |
| 2 | 18-21 | GB4462-6 | 0.59 | -0.23 | 22.8 | 2.738 | 0.28 | - | 0.59 |
| 3 | 0-2 | GB4462-3 | 0.54 | -0.27 | 20.2 | 2.41 | 0.37 | 0.53 | 0.54 |
| 3 | 2-4 | GB4462-3 | 0.53 | -0.27 | 19.8 | 2.62 | 0.27 | 0.49 | 0.53 |
| 3 | 4-6 | GB4462-3 | 0.53 | -0.27 | 19.9 | 2.53 | 0.31 | 0.56 | 0.53 |
| 3 | 6-8 | GB4462-3 | 0.54 | -0.27 | 20.3 | 2.50 | 0.33 | 0.54 | 0.54 |
| 3 | 8-10 | GB4462-3 | 0.53 | -0.27 | 19.9 | 2.54 | 0.31 | 0.61 | 0.53 |
| 3 | 10-12 | GB4462-3 | 0.54 | -0.27 | 20.3 | 2.64 | 0.27 | 0.74 | 0.54 |
| 3 | 12-15 | GB4462-3 | 0.56 | -0.25 | 21.5 | 2.56 | 0.30 | 0.69 | 0.56 |
| 3 | 15-18 | GB4462-3 | 0.55 | -0.26 | 20.9 | 2.77 | 0.26 | 0.74 | 0.55 |
| 3 | 18-21 | GB4462-3 | 0.57 | -0.25 | 21.6 | 2.68 | 0.29 | 0.66 | 0.57 |
| 4 | 0-2 | GB4462-8 | 0.54 | -0.27 | 20.4 | 2.43 | 0.35 | 0.54 | 0.54 |
| 4 | 2-4 | GB4462-8 | 0.53 | -0.27 | 20.0 | 2.59 | 0.30 | 0.37 | 0.53 |
| 4 | 4-6 | GB4462-8 | 0.54 | -0.27 | 20.2 | 2.55 | 0.31 | 0.43 | 0.54 |
| 4 | 6-8 | GB4462-8 | 0.52 | -0.28 | 19.3 | 2.55 | 0.29 | 0.45 | 0.52 |
| 4 | 8-10 | GB4462-8 | 0.53 | -0.27 | 19.9 | 2.69 | 0.26 | - | 0.53 |
| 4 | 10-12 | GB4462-8 | 0.53 | -0.27 | 19.8 | 2.54 | 0.30 | - | 0.53 |
| 4 | 12-15 | GB4462-8 | 0.53 | -0.28 | 19.7 | 2.90 | 0.20 | - | 0.53 |
| 4 | 15-18 | GB4462-8 | - | - | - | - | - | - | - |
| 4 | 18-21 | GB4462-8 | - | - | - | - | - | - | - |

[a] Collected core numbers are relabelled in the sample name to reflect a relative transect position (1-4).

[b] Sediment lithology based on freeze-dried sediments.

[c] TEX$_{86}$ = (GDGT-2 + GDGT-3 + GDGT-5')/(GDGT-1 + GDGT-2 + GDGT-3 + GDGT-5'), (Schouten et al., 2002) applied to both core GDGTs and 1-glycosyl-GDGTs. (1)

[d] TEX$_{86}^{H}$ = log ((GDGT-2 + GDGT-3 + GDGT-5')/(GDGT-1 + GDGT-2 + GDGT-3 + GDGT-5')), for sediments outside the polar regions (Kim et al., 2010).

[e] Following the mean annual sea surface calibration of 0 m water depth (SST = 68.4 × TEX$_{86}^{H}$ + 38.6) of Kim et al. (2010).

[f] Ring index (RI) = 0×(GDGT-0) + 1×(GDGT-1) + 2×(GDGT-2) + 3×(GDGT-3) + 4×(GDGT-4) + 5×(GDGT-5)/ ΣGDGTs, adapted from Pearson et al. (2004). (2)

[g] Methane index (MI) = (GDGT-1 + GDGT-2 + GDGT-3)/(GDGT-1 + GDGT-2 + GDGT-3 + GDGT-5 + GDGT-5') by Zhang et al. (2011).

## 2.2. Lipid extraction





Samples were spiked with a recovery standard (1-alkyl-2-acetoyl-*sn*-glycero-3-
phosphocholine (PAF); Avanti Polar Lipids, Inc.) and extracted using a modified Bligh and
Dyer protocol after Sturt et al. (2004). The extraction involved six steps using 3 different
solvent mixtures. The first four steps involved solvent mixtures of
methanol/dichloromethane/buffer [2:1:0.8; v/v]. From this, the first two steps used a
phosphate buffer (5.5 g/L $Na_2HPO_4$; Avantor Performance Materials, LLC.) adjusted to pH
of 7.4 with HCl; Anachemia Co.), while the third and fourth steps employed a
trichloroacetic acid buffer (50 g/L $C_2HCl_3O_2$; Avantor Performance Materials, LLC. (pH of
2). The final two steps used a solvent mixture of methanol/dichloromethane [5:1; v/v]. Each
extraction step consisted of a 6 ml of solvent mixture, sonicated for 5 min. and centrifuged
for 5 min. at 1250 rpm. After each extraction step, the solvent was decanted and combined
in a separation funnel. The combined extract was purified with milliQ water, heated at ca.
60 °C, and evaporated to dryness under a gentle steam of dry nitrogen. The resulting total
lipid extract (TLE) was then spiked with 1, 2-diheneicosanoyl-*sn*-glycero-3-
phosphocholine ($C_{21}$-PC; Avanti Polar Lipids, Inc.) and subsequently stored at -20 °C
before it was injected for mass spectral analysis.
**2.3. High performance liquid chromatography – mass spectrometry (HPLC-MS)**
A reverse phase electrospray ionization method with a scan range from 100–3000 *m/z* was
chosen for its ability to simultaneous resolve archaeal IPLs and CLs. An aliquot of each
sample representing 1% of the TLE was analyzed using an Agilent Technologies 1260
Infinity II HPLC coupled to an Agilent Technologies 6530 quadruple time-of-flight mass
spectrometer (qToF-MS). Separation was achieved following the method described by Zhu
et al. (2013) using an Agilent Technologies ZORBAX RRHD Eclipse Plus $C_{18}$ (2.1 mm ×
150 mm × 1.8 µm) reverse phase column, fitted with a guard column and maintained at 45
°C. The flow rate was set to 0.25 mL/min. and the gradients were: mobile phase A
(methanol/formic acid/ammonium hydroxide [100:0.04:0.10] v:v:v) held at 100%  for 10
min., followed by a linear gradient to 24% mixing with mobile phase B (propan-2-ol/formic
acid/ammonium hydroxide [100:0.04:0.10] v:v:v) extending for 5 min., a linear gradient to
65% B for 75 min., followed by 70% B for 15 min., that finished by re-equilibrating with
100% A for 15 min. The injection solvent was methanol.
Analyte identification was achieved by accurate mass resolution, mass spectral analysis
using Agilent Technology's MassHunter software and by comparison of fragmentation
patterns with the literature (e.g., Knappy et al., 2009; Liu et al., 2010; Yoshinaga et al.,
2011). Quantification was achieved by summing the integration of peak areas of adducts
$[M+H]^+$, $[M+NH_4]^+$, and $[M+Na]^+$ for the respective GDGTs of interest. The signals for
these compounds were monitored as $[M+H]^+$ on the *m/z* 1464.38, 1462.36, 1460.34,
1458.33 1456.31, 1454.30 mass chromatograms. Additionally, mass fragments consistent
with the loss of a biphytane (*m/z* 743.7) were observed. Once the integrated peak areas were
determined for each GDGT, concentration values were obtained relative to the internal $C_{21}$-
PC standard and reported in µg/g dry sediment weight.
Response factors were determined by a series of injections of a standard solution
containing; 1,2-diacyl-3-O-(α-D-galactosyl1-6)-β-D-galactosyl-*sn*-glycerol (DGDG), 1,2-





diacyl-3-O-β-D-galactosyl-*sn*-glycerol        (MGDG),      1-alkyl-2-acetoyl-*sn*-glycero-3-
phosphocholine    (PAF),    1,2-di-O-phytanyl-*sn*-glycerol    (Archaeol),    1',3'-bis[1,2-
dimyristoyl-*sn*-glycero-3-phospho]-glycerol (14:0 Cardiolipin), 1,2-diheneicosanoyl-*sn*-
glycero-3-phosphocholine ($C_{21}$-PC) from Avanti Polar Lipids, Inc., USA, and  2,2´-di-O-
decyl-3,3´-di-O-(1´´,ω´´-eicosanyl)-1,1´-di-(rac-glycerol)   ($C_{46}$-GTGT)   from   Pandion
Laboratories, LLC in amounts ranging from 100 pg to 30 ng. Concentrations of the standard
mix were then calculated from peak areas of molecular ions in mass chromatograms.
Response factors were calculated relative to the $C_{21}$-PC, and the appropriate correction
factor was applied to the particular lipid class of interest.
A series of samples were re-run to identify or confirm deviations in the data set. The
variations between the concentrations of GDGTs in the re-run and the initial runs yielded a
maximum difference of $\sim \pm 4$ µg/g per GDGT compound, providing confidence in the initial
results and confirming the presence of two outlies in the data set. These outliers are Core 4
at 8-10 cm, with abnormally low concentrations of all compounds that is likely ion
suppression from a sample heavily impregnated with oil, and Core 3 at 15–18 cm, which
contains relatively high lipid concentrations that are yet to be explained.
**3. Results and Discussion**
**3.1. Archaeal lipid diversity and heterotrophic loss**
The Cathedral Hill transect sediments have *i*GDGTs containing 0–4 cyclopentyl (GDGT
0–4) as well as crenarchaeol (Cren) and the isomer of crenarchaeol (Cren') that contains
five rings (four cyclopentyl and one cyclohexyl moiety) (Table S1). Branched GDGTs
include 1a-c, 2a-c, and 3a were found to have discontinuous and/or low absolute
abundances, with some compound classes not being detected (i.e. *br*GDGT-3b). The
*br*GDGTs are therefore not further examined in this study. For cores 1 to 3 the
concentrations of all *i*GDGT compounds systematically decrease with depth (Figure 2).
Bentley et al. (2021) established the sedimentation of archaeal lipids from the upper water
column as being uniform both in terms of spatial loading across the length of the transect
as well as over the past 52.5–105 yrs. of sedimentation penetrated by the length of the push
core. From this, it is estimated that $\sim 70.57 \pm 23.5$ µg *i*GDGTs/g sed./yr. is being deposited
on the seafloor from the upper water column. However, for cores closest to the vent site,
lipid abundances exhibited a much sharper decrease with depth, which Bentley et al. (2021)
attribute to the turnover of archaeal lipids coupled to, but not directly caused by,
hydrothermalism. For cores 1 and 2, losses reach as high as 94% within the upper 21 cmbsf
(cm below sea floor). The lipid loss is less severe for core 3 at ~60%. For the ambient core
4, *i*GDGTs have similar down core stratigraphic trends with a near-consistent average of
400 µg/g sediment concentration and no systematic loss of lipids.
Due to the extreme vent conditions at Cathedral Hill, the identified archaeal *i*GDGT-based
IPLs within the sediments most likely represent the composition of cellular membrane
material from active archaeal communities residing in the sediments. These lipids have
exclusively monoglycosyl (1G) or diglycosyl (2G) head groups linked to a 2,3-sn-glycerol.
Within the pyrolytic environment the transformation of IPL *i*GDGTs could hypothetically





add to the core $i$GDGT lipid pool. Similar to CLs, the 1G-GDGTs range from -0 to -4 and
include Cren and Cren'. Surface concentrations of these lipids are ~15 µg/g sed. in cores 1
to 3 (residing within the microbial mat) and 11 µg/g sed. for core 4 (Table S2). Also similar
to the CLs, the archaeal IPL concentrations decrease down core and are tightly controlled
by porewater temperatures (Table S2).  For cores 1 and 2 the maximum depths for
detectable 1G-GDGTs are 15–18 and 12–15 cmbsf, corresponding to vent porewater
temperatures of 145 and 87 °C, respectively. In core 3, 1G-GDGTs persists down core with
a consistent lipid depletion that reaches its lowest concentration of 5.22 µg/g sed. in the
bottom of the core at 18–21 cmbsf sediment depth where porewater temperatures rise to 80
°C. In core 4, which is most similar to the ambient ocean bottom conditions and falls outside
of the area covered by the microbial mat, the lipid concentrations average is ~8 µg/g sed.
across the depth of the core. The 2G-GDGTs have 0 to 2 cyclopentyl rings that for cores 1
and 2 are restricted to the upper 4 to 6 cmbsf. These lipids are not further investigated in
this study as 2G-GDGTs are of limited abundance (max summed concentrations <7 µg/g
sed.) and their structural diversities negligibly effect isoprenoid-based proxies.
Lipid-based proxies for the calibration or reconstruction of paleoclimate records such as
$TEX_{86}$, BIT, CBT, and MBT, are based on environmentally scaled loadings of select GDGT
compound classes. These proxies could be negatively impacted should other ocean floor
sediment systems experience high rates of lipid turnover (Lengger et al., 2014). To evaluate
whether down-core depletions of lipid concentrations impacted tetraether-based proxies,
the concentrations of the highly abundant GDGT-0 was plotted relative to the $TEX_{86}$ ratio
lipids ($i$GDGT-1, -2, -3, and Cren') (Figure 3A). For figure 3A, straight lines in the
logarithmic plot indicate near-equal depletion rates between the paired x- and y-axis lipid
classes. Similarly, parallel lines between lipid pairs also indicates near-equal depletion
rates, with vertical offsets between pairs marking different initial starting abundances
between the paired lipid classes. In this regard, $i$GDGT-0, -1, -2, and Cren' have undergone
the same rate of turnover. However, the depletion rate of $i$GDGT-3's is lower than that of
other lipid classes for cores 1 and 2. Although, this may represent a distinct resilience to
turnover, we suggest it results from overprinting by the subsurface hyperthermophilic
archaeal community (see below).
To better track changes across each core, the degradation rate constants (k') of $TEX_{86}$ lipid
classes were calculated for each push core (Figure A2; Table A3) using a first-order kinetic
model:
$C_t = C_i \cdot e^{-k't}$                              (5)
in which $C_t$ and $C_i$ are concentration at time ($t$) and the initial concentration, respectively
(Schouten et al., 2010).  Rearranging Eq. 5, the k' were calculated as
$k' = (-\ln[C_t/ C_i])/t$                              (6)

From these data, it is evident that the down core concentrations of each lipid decrease at
equivalent rates for all but core 2 (i.e. they have the same slopes for their rates of decay;





m$_{\log k'}$). This is consistent with the TEX$_{86}$ $i$GDGT lipid classes being removed from the
sediment lipid pool in a non-selective manner.
Lastly, TEX$_{86}$, RI, and MI values were plotted against their respective summed $i$GDGTs
lipid concentrations (Fig 3B–D). For samples located within the habitable zone (having
porewaters ranging from 0–123˚C; Kashefi and Lovley, 2003), no correlation is observed
between the lipid abundances and proxy ratios of TEX$_{86}$, RI, or MI (Figure 3B–D). This
further suggests these proxies are not affected by turnover in the habitable zone. However,
once sediment burial reaches beyond the habitable zone, TEX$_{86}$ ratios trend to higher values
(similarly also reflected in GDGT-3 concentration trends of Figure 3A). Collectively, these
data strongly indicate that archaeal lipid turnover is largely nonselective of the TEX$_{86}$ lipid
classes and will therefore theoretically not in and of itself significantly impact archaeal
lipid paleoclimate proxy reconstructions.

Apart from paleoclimate reconstructions, the archaeal lipid data can also be used to resolve
some aspects of the local biogeochemical cycles present at the vent site. Maximal anaerobic
oxidation of methane (AOM) at Guaymas Basin has been observed at 35 to 90 °C, but
generally accounts for less than 5% of sulfate reduction (Kallmeyer and Boetius, 2004).
For example, highly $^{13}$C-depleted CLs reaching up -70‰ in hydrothermal vent sediments
with porewater temperatures as high as 95 °C indicates thermophilic archaea actively
engaging in AOM (Schouten et al., 2003). The methane index (MI; Table 1) can be used to
differentiate regions of normal marine (values between 0–0.3) and active AOM conditions
where values >0.5–1 for gas hydrate impacted sediments and subsurface environments with
high levels of AOM (Stadnitskaia et al., 2008, Zhang et al., 2011). When applying the MI
to the Cathedral Hill sediments very low values are recorded with no correspondence to
thermal controls. Although, it could be considered that this arises from selective
degradation; the very low MI values are equally explained by broad loading of $i$GDGTs
from the upper water column. As such, the low AOM activities may also indicate microbial
ammonia oxidation, which has been shown to influence the TEX$_{86}$ proxy (Hurley et al.,
20016) is likely not a significant factor in this setting.



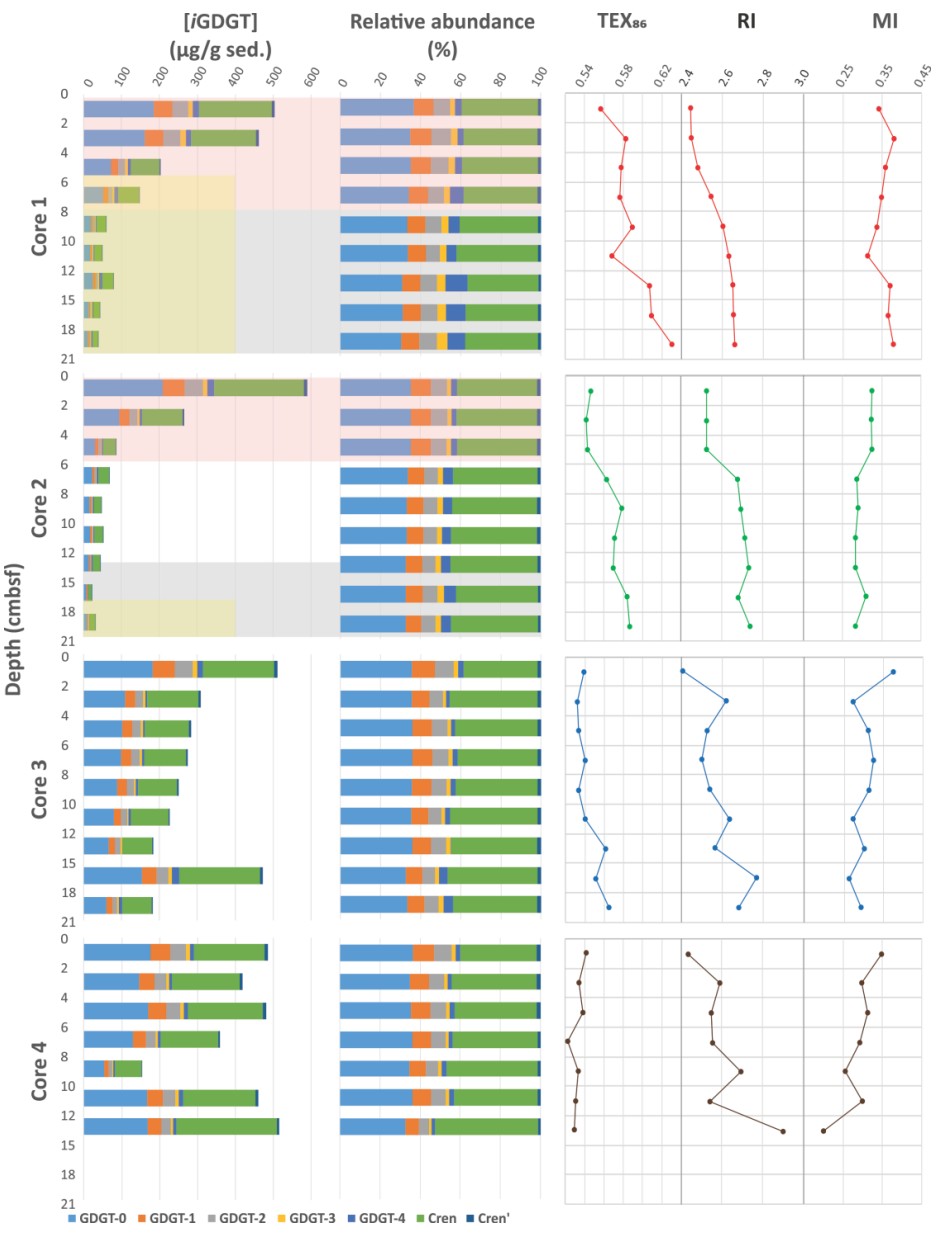

**FIGURE 2.** Down core profiles of the Cathedral Hill core *i*GDGTs absolute and relative lipid abundances and their generated *i*GDGT proxies: TEX$_{86}$, RI, and MI. Pink regions indicate transect intervals within zones of active GDGT lipid heterotrophy (Bentley et al., 2021). Grey regions mark regions where porewater temperatures exceed 123 °C marking a zone beyond the upper thermal limit of life. Yellow fields indicate regions where oil generation and hydrocarbon degradation has been noted to occur (Dalzell et al., 2021).






•Core 1 •Core 2 •Core 3 •Core 4

**FIGURE 3.** A) Comparison of TEX$_{86}$ lipid concentrations GDGT-1 (circles), -2 (squares),
-3 (triangles), and Cren' (diamonds) relative to the GDGT-0. Comparison of B) TEX$_{86}$, C)
RI, and D) MI proxy values relative to summed $i$GDGTs abundances of the Cathedral Hill
transect cores. Light green and pink regions indicate areas within and outside the habitable
zone of life. Solid and dotted regression lines mark the total number of samples investigated
for this study ($n$=34) and those that only reside within the habitable zone where up to 94%
of the archaeal lipid turnover occurs ($n$=22), respectively.




### 3.2. TEX$_{86}$ and reconstructed SSTs

McClymont et al. (2012) reported a GDGT-based reconstructed annual SSTs of 16–18 °C for ambient sediment in the Guaymas Basin during an annual cycle from 1996–1997 following the calibration model for sediments outside of polar regions proposed by Kim et al. (2010). These authors demonstrated the temperatures derived from the TEX$_{86}$ reconstruction were significantly lower than those derived from the closely co-varying U$_{37}^{k\prime}$, an alkenone lipid-based paleoclimate proxy (Brassell et al., 1986), and satellite measured estimates that jointly produced a mean annual sea surface temperature (MASST) of 23 °C. The longer 21-year (1982–2004) satellite-derived MASST is also reported to be higher at 24 °C (Herrera-Cervantes et al., 2007). Although the sites and time frames of these surveys do not match that of the Cathedral Hill survey, they do provide context to what our reconstructed TEX$_{86}$ values should record.

The high sedimentation rate at Cathedral Hill has resulted in near homogenous inputs of organic matter from the upper water column across the transect area (Dalzell et al., 2021; Bentley et al., 2021). Therefore, TEX$_{86}$ reconstructions should produce equivalent cross-transect trends with sediment depth. Nonetheless, as with changes in the archaeal lipid concentrations, the profiles of $i$GDGT proxies TEX$_{86}$ and RI of the transect similarly have down core trends (Figure 2; Bentley et al., 2021). For core 4, TEX$_{86}$ span a narrow range of values (n=7; 0.52–0.54, avg. 0.53 ± 0.01; Figure 4A) across a period of ~ 37.5 to 75 yrs. To a slightly lesser degree, the shallow-surface samples (0-2 cmbsf) across the transect also display near-equal values to core 4 (n=4; 0.56–0.54; avg. 0.55 ± 0.01). These values mark a TEX$_{86}^{H}$ reconstructed mean annual SST of 19.3–20.4 °C following the Kim et al. (2010) calibration model (Table 1). However, the TEX$_{86}$ values recorded in cores 1 to 3 at Cathedral Hill have considerably larger ranges that systematically increase with rising porewater temperatures (R$^2$ = 0.83; Table 1; Figure 2 and 4A). This increase is most noticeable in core 1 where the highest TEX$_{86}$ values are obtained from the bottom core sediments (10–21 cmbsf) where TEX$_{86}$ values span 0.57–0.63 (Table 1; Fig 4A) corresponding to a TEX$_{86}^{H}$ reconstructed SST change of 3.1 °C marking a range from 21.8 to 24.9 °C (Table 1). Since the Cathedral Hill transect only spans ~8 m, the fundamental driver for the proxy's increases must be exposure to *in situ* vent fluid temperatures (Figure 4).

Two mechanisms are considered for the observed proxy variations. The first is that progressive ring-loss due to carbon-carbon bond cleavage of pentacyclic rings moieties by exposure to the sharp geothermal gradient at Cathedral Hill acts to systematically attenuate the $i$GDGT lipid pool. Hydrous pyrolysis experiments conducted by Schouten et al. (2004) demonstrated that at extreme temperatures (ca. >160 °C), TEX$_{86}$ values become negatively impacted by the preferential destruction of polycyclic GDGTs. Such losses produce progressively lower ratio values. Although, the transect sediment porewaters do not reach the pyrolytic temperatures of the Schouten et al. (2004) experiment, they are high enough to generate hydrocarbons (Dalzell et al., 2021) and thermochemically degrade $i$GDGTs in the hottest regions of the transect. However, the observed stratigraphic TEX$_{86}$ trends do not match those of predicted ring loss as the values increase rather than decrease in relation to



elevated porewater condition. Nonetheless, the thermochemical oxidative loss of GDGTs
and its effect on the TEX$_{86}$ ratio is further explored below (section 3.4).

The second mechanism is that subsurface microbial communities donate enough core
GDGTs to overprint the detrital signal source. The RI (Figure 4B) values were similarly
compared to recorded porewater temperatures to better interpret the TEX$_{86}$ trends and to
ensure that the Cathedral Hill reconstructed temperatures are influenced by the subsurface
microbial community. In this regard, RI is used to monitor the adaptive response of an
archaeal community at the hydrothermal vent site. Lipid cyclization is an adaptive response
to changing environmental temperature or acidity in which an archaeon increases its
rigidity by decreasing the fluidity and permeability of its cellular membrane that, therefore,
also further regulates the flow of solutes and nutrients in and out of the cell (Gliozzi et al.,
1983; De Rosa and Gambacorta, 1988; Uda et al., 2001; Schouten et al., 2002; Macalady
et al., 2004; Boyd et al., 2013). Both cores 1 and 2 have RI values highly correlated to
temperature ($R^2 = 0.87$ and 0.75, respectively) consistent with heat stress adaption. As such,
a significant proportion of the measured *i*GDGTs likely emanate from archaeal
communities living in the shallow sediments of Cathedral Hill. In this regard, the lipid
cyclization pattern may reflect stratigraphically discrete thermophilic to hyperthermophilic
communities that are selectively adapted to more extreme temperature conditions.


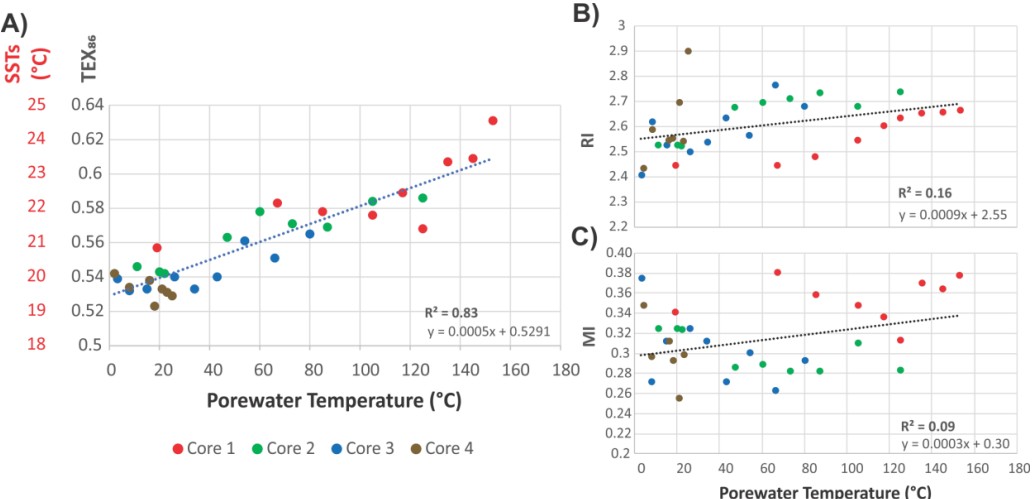

**FIGURE 4.** Cross plots of A) TEX$_{86}$, B) RI, and C) MI, *i*GDGT proxies versus porewater
temperature. TEX$_{86}^{H}$ reconstructed MASSTs are based on Kim et al. (2010).

**3.3. Lipid signal sourcing**
To evaluate the sources of measured archaeal lipids, core and $_{IPL}$TEX$_{86}$ indices were
compared as signal response loadings from their respective pools of living and dead cellular





debris (Figure 5). For cores 1, 2, and 3 the 1G-*i*GDGT $_{IPL}$TEX$_{86}$ measures are correlated
with temperature ($R^2$ = 0.46, 0.74, and 0.66, respectively; Figure 5A). In this regard, 1G-
*i*GDGT $_{IPL}$TEX$_{86}$ ratio appears to also measure *in situ* porewater temperatures.  Factors
such as community composition and adaptation may further impact the $_{IPL}$TEX$_{86}$ ratio as
the rates of changes between cores 1–3 are not the same. Similar to the CL TEX$_{86}$ values,
the $_{IPL}$TEX$_{86}$ are not correlated to their summed TEX$_{86}$ lipid abundances (Figure 5B). Such
a condition is consistent with the living lipid pool being modified by the archaeal
community's response to thermal stress and not by subsequent thermal oxidative
transformation occurring shortly after cell death.
The IPL and CL lipids of transect samples can be further grouped into three clusters (A, B,
C), suggesting a mixed signal for the sourcing of archaeal GDGTs from both the living and
dead pools of archaea (Figure 5C). In this plot, we assume that clusters falling on the 1:1
line indicate the living biota can equally contribute to the dead pool of total recovered
GDGTs.  Those off-axis contribute either less or more to one or the other lipid pool. The
three clusters mark unique thermal zones within the transect area with cluster A being
composed of the ambient core 2 to 4 seafloor surface samples; cluster B marking a mix of
intermediate temperature samples from all cores; and cluster C being composed of high
temperatures samples. The lipid groups likely mark distinct archaeal communities. As
cluster B resides on the 1:1 line, the TEX$_{86}$ core lipids likely have a mixed of detrital and
*in situ* inputs. Cluster C, however, appears likely dominated by *in situ* lipid production. The
hyperthermophilic *Methanopyrus kandleri,* recovered from other Guaymas Basin sites
(Teske et al., 2014), may represent one such archaeon contributing to the cluster C lipid
pool. The thermal zonation and equivalent directionality of the resulting ratios (i.e. both CL
and IPL TEX$_{86}$ ratios increase with porewater temperature) further supports overprinting
of the original CL TEX$_{86}$ sea surface signal by the ocean bottom sediment archaeal
community as a mechanism for the observed CL TEX$_{86}$ trends.
Collectively, these results suggest the source of the archaeal core lipids measured in the
TEX$_{86}$ and RI indices progressively become more dominated by subsurface microbial
communities adapted to the hotter hydrothermal vent fluids. Our results also indicate that
in select natural environments, such as hydrothermal vent complexes, the TEX$_{86}$ SST-proxy
may entirely record ocean bottom sediment porewater temperatures. To our knowledge, a
clear case of overprinting to this level has not yet been demonstrated.



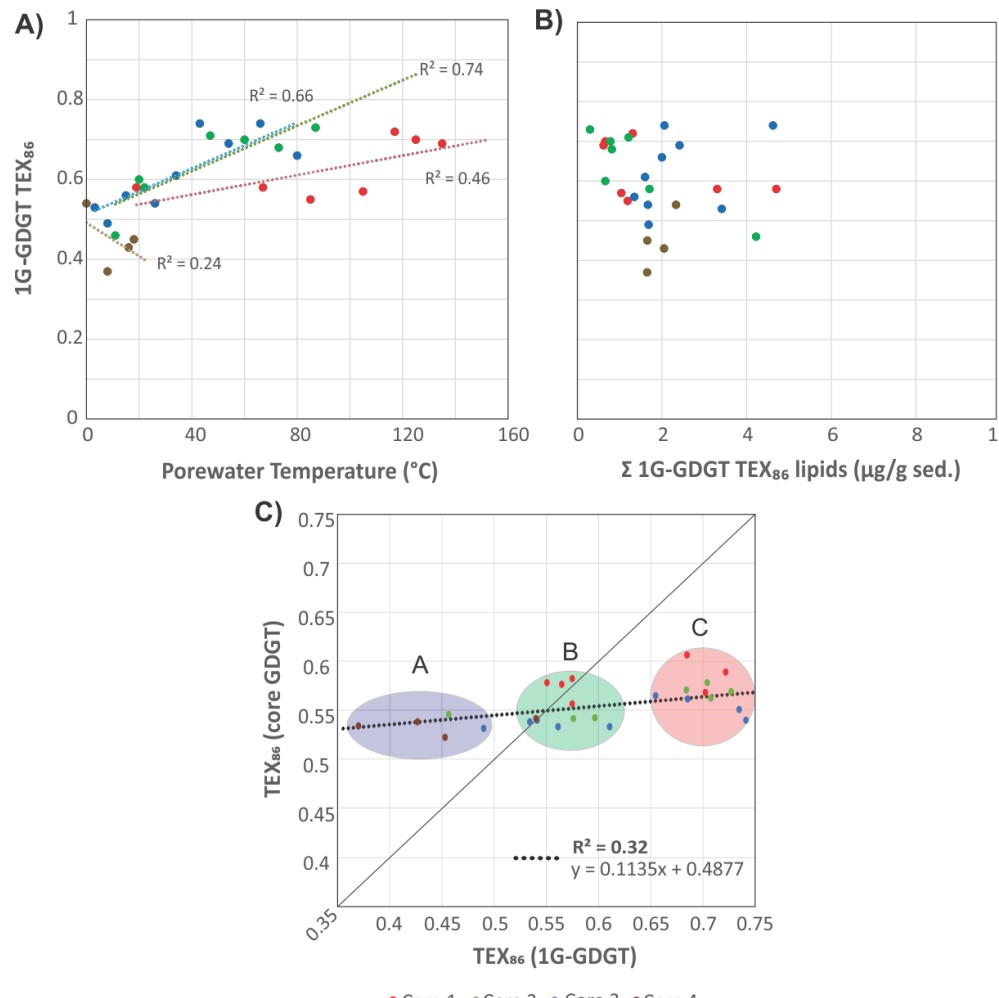


**FIGURE 5.** Cross plots of 1G-*i*GDGTs IPLTEX86 versus (A) porewater temperatures and (B) the concentration of 1G-*i*GDGTs in the sediments. C) TEX86 proxy of core GDGTs vs 1G-GDGTs. Clusters A–C may represent differernt archeal communities that are providing varying inputs of *i*GDGT to the core GDGT lipid pool. The dotted trendline is the particial least square regression of the complete core lipid TEX86 data set. The solid line marks the 1:1 CL to IPL proxy correspondance indicating both allochthonous and autochthonous sources contribute equally to the core GDGT lipid pool.






### 3.4. TEX$_{86}$ overprint corrections

The measured TEX$_{86}$ ($_M$TEX$_{86}$) value of the Cathedral Hill sediments is herein considered
to be a weighted sum of a sea surface TEX$_{86}$ ($_{SS}$TEX$_{86}$) value acquired from lipids sourced
in the upper water column that is further modified by a component of water column sourced
core lipids ($_{wC}$TEX$_{86}$) as well as by additions of archaeal lipids from the benthic and
subsurface microbial communities ($_{Sed}$TEX$_{86}$). These ratio loadings are potentially further
modified by diagenetic influences in the ocean bottom sediments. Over the cumulative
sediment burial period and measured porewater temperatures of the Cathedral Hill push
core sediments, these influences include the selective loss of lipids by their binding into
protokerogen ($K$) and potential changes due to the loss of lipid by turnover ($\varphi$; section 3.1).
Additional catagenetic effects from thermochemical alteration of lipids ($\theta$) may also
attenuate the sum of sedimentary core lipids by their exposure to high temperature vent
fluids. Collectively, these effects are considered to form the following relationship:

$$_M\text{TEX}_{86} = \frac{a_{SS}\text{TEX}_{86} + b_{wC}\text{TEX}_{86} + c(d_{0-n})_{Sed}\text{TEX}_{86}}{\varphi + K + \theta}$$ (7)

where $a$, $b$, and $c$, are measured scaling parameters for lipid loading and $\varphi$, $K$, and $\theta$ are
diagenetic and catagenetic alteration parameters. Solving for $_{SS}$TEX$_{86}$:

$$_{SS}\text{TEX}_{86} = \frac{_M\text{TEX}_{86}(\varphi + K + \theta)}{a} - \frac{b_{wC}\text{TEX}_{86} + c(d_{0-n})_{Sed}\text{TEX}_{86}}{a}$$ (8)

In this regard, a portion of the archaeal community from the deeper water column,
presumably initially sourced of IPLs, and an additional community inhabiting the ocean
floor sediments are assumed to eventually die with their respective IPLs gradually
becoming converted to CLs that further contribute to the observed $_M$TEX$_{86}$ value. For this
study, no data were collected to calculate $b_{wC}$TEX$_{86}$ and its potential impact on $_M$TEX$_{86}$ is
not further considered. However, it is highly likely, given the longer residence times for
glycosidic-based headgroups of the identified archaeal IPLs and their relatively short
settling time through the water column that a component of this lipid source could already
be mixed with the $_{Sed}$TEX$_{86}$ value (Lengger et al., 2012). For this study, $_{Sed}$TEX$_{86}$ is an IPL-
TEX$_{86}$ ratio based on detected 1G-GDGT-1, -2, -3, and Cren' as present in the original
paleoclimate proxy (Eq. 1; Table 1; Figure 6). The 2G-GDGT lipids are excluded from the
calculation due to their low absolute concentrations (<2 µg/g sed.), their limited number of
detected TEX$_{86}$ core lipid configurations (comprising only of GDGT-1 and GDGT-2; Table
A2), and their short stratigraphic zones of occurrence (section 3.1). The $_{Sed}$TEX$_{86}$ is further
scaled by the summed concentrations of these lipids as they increasingly accumulate with
sediment depth ($d_{0-n}$). For Cathedral Hill, the sum of allochthonous TEX$_{86}$ lipids ($\Sigma$[GDGTs
$_{CL\text{-}TEX_{86}}$ lipids]$_{0-2}$) is estimated to be 120 µg/g sed. based on an average surface lipid
concentration (0-2 cmbsf) measured across the four core transect. As such,

$$c(d_{0-n}) = \sum_{i=0}^{n} \left( \frac{[\text{GDGTs}_{IPL\text{-}TEX_{86}\text{ lipids}}]_n}{[\text{GDGTs}_{CL\text{-}TEX_{86}\text{ lipids}}]_{0-2cm}} \right)$$ (9)





where $n$ is the deepest point of sediment burial.

Selective lipid removal by digenetic and catagenetic processes theoretically may also affect
the $TEX_{86}$ value; however, their perspective impact on the directionality and magnitude of
the ratio are difficult to predict and equally hard to discretely measure. For Cathedral Hill,
although the loss of GDGTs to protokerogen formation could potentially impact the ratio,
it has been proven to be very low for the analyze sediments (Bentley et al., 2021). As such,
the selectivity of lipid classes being adsorbed to a protokerogen is undeterminable. More
importantly, for this site it is insignificant, and the $K$ parameter in Eqs. 7 and 8 is therefore
assigned a value of 0.

The degradation rates of each $TEX_{86}$ lipid class were independently measured for the four
push cores (Eq. 6; Fig. A2). Given the high geothermal gradient at Cathedral Hill, some of
the transect push core sediments resided within zones of active catagenesis (Fig. 2; Dalzell
et al., 2021). As the abundance of both CLs and IPLs differentially decreases through the
various core sediment profiles with turnover rates that appear to be constrained by
porewater temperature changes (section 3.1), the degradation rates must also record the
effects of thermochemical oxidative weathering (Fig. 3B). In this case, $\varphi$ and $\theta$ are
therefore treated as a grouped parameter.

To determine if individual lipid classes were selectively removed during degradation, the
variance ($s^2$) of the rate change as measured from its respective regression slope (i.e. $m_{logk'}$)
of the $TEX_{86}$ lipid classes (Fig. A2; Supplemental Table A3 from Eq. 6) were calculated.
For the Cathedral Hill transect, the calculated $m_{logk'}$ $s^2$ is 0.11, which suggests near equal
degradation rates for all $TEX_{86}$ lipid classes. Therefore, lipid turnover and the concomitant
thermochemical oxidation of these lipid classes is also similarly non-selective. A weighing
function for the degree of lipid class selectivity during turnover is nonetheless proposed:

$$\varphi + \theta = 1 / {_M}TEX_{86}{}^{0.11} \qquad (10)$$

When applied to Eq. 8 minor changes to the reconstructed lumped $_{SS+WC}TEX_{86}$ ratio are
observed consistent with the absence of a comparative relationship between $i$GDGT down
core lipid depletions and the respective $_MTEX_{86}$ ratios across the biologically active zone
of the transect sediments (section 3.1; Figure 3B).

Equation 7 predicts an average transect $_{SS+WC}TEX_{86}^{H}$ reconstructed SST of 21.92 ±0.66 °C
with no elevated trends for increasing porewater temperatures across each of the transect
cores (Table 2; Figure 6A). If the $\varphi$, $K$, and $\theta$ scaling parameters are removed from the
calculation the average temperature shifts 2.08 °C lower to 19.69 ±0.39 °C (Table 2; Figure
6B). The marginal change is likely due to only a few sediment samples displaying evidence
of $in\ situ$ hydrocarbon generation associated with thermochemical oxidation (Dalzell et al.,
2021). Irrespective of approach, but particularly the case for the more simplified
expression, all measures produce values closer to the expected SST of 19.3–20.4 °C that is
based on the range of values recorded for core 4 and the three transect surface sediments
(section 3.2). These values are ~3 °C lower than the 23–24 °C obtained for the 21-year
(1982–2004) satellite-derived MASST data for the Guaymas Basin region (Herrera-



Cervantes et al., 2007). Nonetheless, nearly all $_M$TEX$_{86}$ attenuation can therefore be
attributed to sediment microbial overprinting. The high degree of influence is striking given
that the upper water flux of GDGTs is estimated to represents up to 93% of the total intact
polar and core GDGT lipid pool within these sediments. In this regard, it demonstrated that
microbial community influences TEX$_{86}$ measurements.

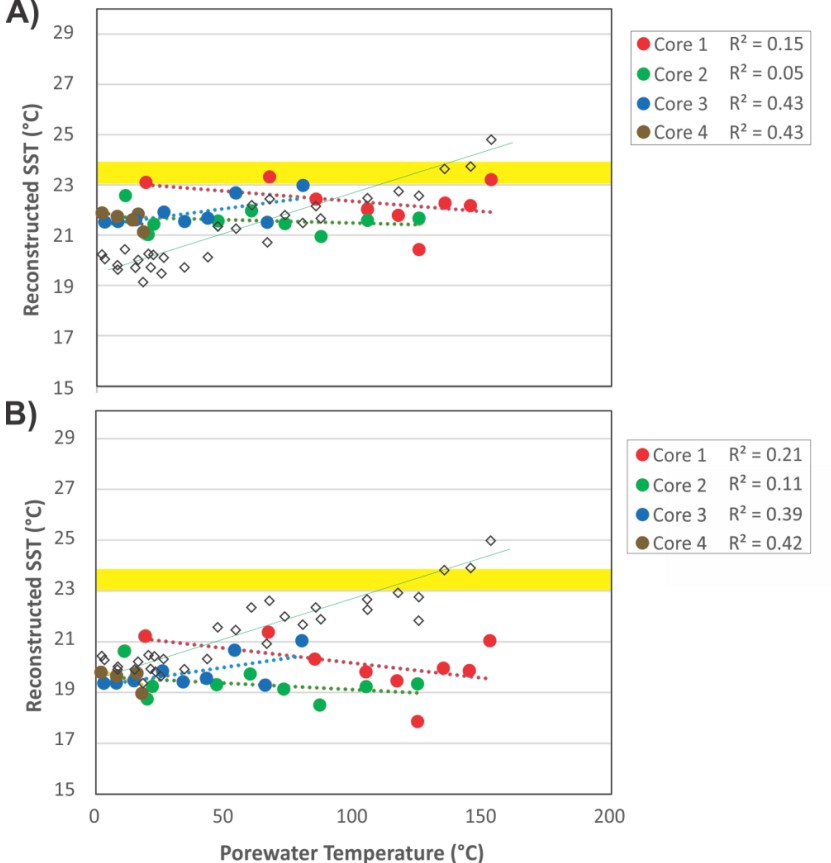

**FIGURE 6.** Reconstructed $_{SS}$TEX$_{86}$ SSTs from (A) Eq. 8 and (B) Eq. 8 without $\varphi$, $K$, and
$\theta$ scaling parameters compared to measured porewater temperatures. $_M$TEX$_{86}$ values are
also plotted for reference (open green circles). Yellow field is the 23–24 °C range observed
for the 21-year (1982–2004) satellite-derived MASST data (Herrera-Cervantes et al.,
2007). The corrected data series show a lack of correlation suggesting that model can back-
out the original SST signal.





**Table 2.** Reconstructed sea surface temperatures.

| Sample | Depth (cmbsf) | Porewater Temp. (°C) | $t$ Time (yrs.) | $_M$TEX$_{86}$ (Measured $i$GDGT TEX$_{86}$) | Reconstructed SST (°C) | TEX$_{86}$ 1G-GDGT IPLs (µg/g) | Cumulative 1G-GDGTs Loading with Depth (µg/g) | $_{Sed}$TEX$_{86}$ (i.e. 1G-GDGT $_{IPL}$TEX$_{86}$) | $c(d_{0-n})$ Cumulative Weighted IPL Loading (Eq. 9) |
|---|---|---|---|---|---|---|---|---|---|
| Core 1 (0-2cm) | 1 | 19 | 10 | 0.56 | 21.2 | 4.80 | 0 | 0.58 | 0.00 |
| Core 1 (2-4cm) | 3 | 67 | 20 | 0.58 | 22.6 | 3.41 | 4.80 | 0.58 | 0.04 |
| Core 1 (4-6cm) | 5 | 85 | 30 | 0.58 | 22.3 | 1.29 | 8.21 | 0.55 | 0.07 |
| Core 1 (6-8cm) | 7 | 105 | 40 | 0.58 | 22.2 | 1.14 | 9.50 | 0.57 | 0.08 |
| Core 1 (8-10cm) | 9 | 117 | 50 | 0.59 | 22.9 | 1.41 | 10.64 | 0.72 | 0.09 |
| Core 1 (10-12cm) | 11 | 125 | 60 | 0.57 | 21.8 | 0.76 | 12.05 | 0.70 | 0.10 |
| Core 1 (12-15cm) | 13 | 135 | 70 | 0.61 | 23.8 | 0.72 | 12.81 | 0.69 | 0.11 |
| Core 1 (15-18cm) | 17 | 145 | 80 | 0.61 | 23.9 | 0.00 | 13.53 | 0.69 | 0.11 |
| Core 1 (18-21cm) | 20 | 153 | 90 | 0.63 | 24.9 | 0.00 | 13.53 | 0.69 | 0.11 |
| **Avg.** | | | | **0.59** | **22.84** | | | | |
| **Std. Dev.** | | | | **0.02** | **1.16** | | | | |
| Core 2 (0-2cm) | 1 | 11 | 10 | 0.55 | 20.6 | 4.33 | 0 | 0.46 | 0.00 |
| Core 2 (2-4cm) | 3 | 22 | 20 | 0.54 | 20.4 | 1.80 | 4.33 | 0.58 | 0.04 |
| Core 2 (4-6cm) | 5 | 20 | 30 | 0.54 | 20.5 | 0.76 | 6.13 | 0.60 | 0.05 |
| Core 2 (6-8cm) | 7 | 47 | 40 | 0.56 | 21.5 | 1.31 | 6.89 | 0.71 | 0.06 |
| Core 2 (8-10cm) | 9 | 60 | 50 | 0.58 | 22.3 | 0.88 | 8.20 | 0.70 | 0.07 |
| Core 2 (10-12cm) | 11 | 73 | 60 | 0.57 | 22.0 | 0.92 | 9.08 | 0.68 | 0.08 |
| Core 2 (12-15cm) | 13 | 87 | 70 | 0.57 | 21.8 | 0.40 | 10.00 | 0.73 | 0.08 |
| Core 2 (15-18cm) | 17 | 105 | 80 | 0.58 | 22.6 | 0.00 | 10.40 | 0.73 | 0.09 |
| Core 2 (18-21cm) | 20 | 125 | 90 | 0.59 | 22.7 | 0.00 | 10.40 | 0.73 | 0.09 |
| **Avg.** | | | | **0.56** | **21.61** | | | | |
| **Std. Dev.** | | | | **0.02** | **0.91** | | | | |
| Core 3 (0-2cm) | 1 | 3.2 | 10 | 0.54 | 20.2 | 3.51 | 0 | 0.53 | 0.03 |
| Core 3 (2-4cm) | 3 | 8 | 20 | 0.53 | 19.9 | 1.79 | 3.51 | 0.49 | 0.01 |
| Core 3 (4-6cm) | 5 | 15 | 30 | 0.53 | 19.9 | 1.45 | 5.30 | 0.56 | 0.01 |
| Core 3 (6-8cm) | 7 | 26 | 40 | 0.54 | 20.3 | 1.77 | 6.74 | 0.54 | 0.01 |
| Core 3 (8-10cm) | 9 | 34 | 50 | 0.53 | 19.9 | 1.70 | 8.51 | 0.61 | 0.01 |
| Core 3 (10-12cm) | 11 | 43 | 60 | 0.54 | 20.3 | 2.16 | 10.21 | 0.74 | 0.02 |
| Core 3 (12-15cm) | 13 | 54 | 70 | 0.56 | 21.4 | 2.52 | 12.37 | 0.69 | 0.02 |
| Core 3 (15-18cm) | 17 | 66 | 80 | 0.55 | 20.9 | 4.72 | 14.89 | 0.74 | 0.04 |
| Core3 (18-21cm) | 20 | 80 | 90 | 0.57 | 21.6 | 2.10 | 19.61 | 0.66 | 0.02 |





| | | | | | | | | | |
|---|---|---|---|---|---|---|---|---|---|
| **Avg.** | | | | **0.54** | **20.50** | | | | |
| **Std. Dev.** | | | | **0.01** | **0.67** | | | | |
| Core 4 (0-2cm) | 1 | 2 | 10 | 0.54 | 20.4 | 2.43 | 0 | 0.54 | 0.02 |
| Core 4 (2-4cm) | 3 | 8 | 20 | 0.53 | 20.0 | 1.75 | 2.43 | 0.37 | 0.01 |
| Core 4 (4-6cm) | 5 | 16 | 30 | 0.54 | 20.2 | 2.15 | 4.18 | 0.43 | 0.02 |
| Core 4 (6-8cm) | 7 | 18 | 40 | 0.52 | 19.3 | 1.76 | 6.34 | 0.45 | 0.01 |
| Core 4 (8-10cm) | 9 | 21 | 50 | 0.53 | 19.9 | 0.44 | 8.09 | - | - |
| Core 4 (10-12cm) | 11 | 23 | 60 | 0.53 | 19.8 | 2.20 | 8.54 | - | - |
| Core 4 (12-15cm) | 13 | 25 | 70 | 0.53 | 19.7 | 0.00 | 10.74 | - | - |
| **Avg.** | | | | **0.53** | **19.90** | | | | |
| **Std. Dev.** | | | | **0.01** | **0.34** | | | | |
| **Avg.** | | | | | | | | | |
| **Std. Dev.** | | | | | | | | | |



**Table 2.** Reconstructed sea surface temperatures (continued).

| Sample | Eq. 8 excluding φ+θ+K | | | Eq. 8 including φ+θ+K | | |
|---|---|---|---|---|---|---|
| | $_{SS+WC}TEX_{86}$ ($_M TEX_{86}$ - $c(d_{0-n})*_{Sed}TEX_{86}$) | $_{SS+WC}TEX_{86}^{H}$ (after Kim et al., 2010) | $_{SS+WC}TEX_{86}^{H}$ Reconstructed SST (°C) | φ+θ (Eq. 10) (where $s^2$ = 0.11; Table A3) | $_{SS+WC}TEX_{86}$ | $_{SS+WC}TEX_{86}^{H}$ Reconstructed SST (°C) (after Kim et al., 2010) |
| Core 1 (0-2cm) | 0.56 | -0.25 | 21.2 | 1.07 | 0.59 | 23.1 |
| Core 1 (2-4cm) | 0.56 | -0.25 | 21.4 | 1.07 | 0.60 | 23.3 |
| Core 1 (4-6cm) | 0.54 | -0.27 | 20.3 | 1.07 | 0.58 | 22.5 |
| Core 1 (6-8cm) | 0.53 | -0.27 | 19.8 | 1.07 | 0.57 | 22.0 |
| Core 1 (8-10cm) | 0.52 | -0.28 | 19.5 | 1.07 | 0.57 | 21.8 |
| Core 1 (10-12cm) | 0.50 | -0.30 | 17.9 | 1.08 | 0.54 | 20.5 |
| Core 1 (12-15cm) | 0.53 | -0.27 | 20.0 | 1.07 | 0.58 | 22.3 |
| Core 1 (15-18cm) | 0.53 | -0.27 | 19.8 | 1.07 | 0.58 | 22.2 |
| Core 1 (18-21cm) | 0.55 | -0.26 | 21.0 | 1.07 | 0.60 | 23.2 |
| **Avg.** | **0.54** | **-0.27** | **20.10** | **1.07** | **0.58** | **22.33** |
| **Std. Dev.** | **0.02** | **0.02** | **1.08** | **0.00** | **0.02** | **0.89** |
| Core 2 (0-2cm) | 0.55 | -0.26 | 20.6 | 1.07 | 0.58 | 22.6 |
| Core 2 (2-4cm) | 0.52 | -0.28 | 19.2 | 1.07 | 0.56 | 21.5 |
| Core 2 (4-6cm) | 0.51 | -0.29 | 18.7 | 1.08 | 0.55 | 21.1 |
| Core 2 (6-8cm) | 0.52 | -0.28 | 19.3 | 1.07 | 0.56 | 21.6 |
| Core 2 (8-10cm) | 0.53 | -0.28 | 19.7 | 1.07 | 0.57 | 22.0 |
| Core 2 (10-12cm) | 0.52 | -0.28 | 19.1 | 1.07 | 0.56 | 21.5 |
| Core 2 (12-15cm) | 0.51 | -0.29 | 18.5 | 1.08 | 0.55 | 21.0 |
| Core 2 (15-18cm) | 0.52 | -0.28 | 19.2 | 1.07 | 0.56 | 21.6 |
| Core 2 (18-21cm) | 0.52 | -0.28 | 19.3 | 1.07 | 0.57 | 21.7 |
| **Avg.** | **0.52** | **-0.28** | **19.32** | **1.07** | **0.56** | **21.61** |
| **Std. Dev.** | **0.01** | **0.01** | **0.60** | **0.00** | **0.01** | **0.49** |
| Core 3 (0-2cm) | 0.52 | -0.28 | 19.4 | 1.07 | 0.56 | 21.5 |
| Core 3 (2-4cm) | 0.52 | -0.28 | 19.4 | 1.07 | 0.56 | 21.6 |
| Core 3 (4-6cm) | 0.53 | -0.28 | 19.5 | 1.07 | 0.57 | 21.7 |
| Core 3 (6-8cm) | 0.53 | -0.27 | 19.9 | 1.07 | 0.57 | 21.9 |
| Core 3 (8-10cm) | 0.52 | -0.28 | 19.4 | 1.07 | 0.56 | 21.6 |
| Core 3 (10-12cm) | 0.53 | -0.28 | 19.6 | 1.07 | 0.57 | 21.7 |
| Core 3 (12-15cm) | 0.55 | -0.26 | 20.7 | 1.07 | 0.59 | 22.7 |
| Core 3 (15-18cm) | 0.52 | -0.28 | 19.3 | 1.07 | 0.56 | 21.5 |
| Core3 (18-21cm) | 0.55 | -0.26 | 21.0 | 1.07 | 0.59 | 23.0 |
| **Avg.** | **0.53** | **-0.27** | **19.79** | **1.07** | **0.57** | **21.91** |
| **Std. Dev.** | **0.01** | **0.01** | **0.62** | **0.00** | **0.01** | **0.55** |
| Core 4 (0-2cm) | 0.53 | -0.27 | 19.8 | 1.07 | 0.57 | 21.9 |
| Core 4 (2-4cm) | 0.53 | -0.28 | 19.7 | 1.07 | 0.57 | 21.8 |





| | | | | | | |
|---|---|---|---|---|---|---|
| Core 4 (4-6cm) | 0.53 | -0.28 | 19.8 | 1.07 | 0.57 | 21.9 |
| Core 4 (6-8cm) | 0.52 | -0.29 | 19.0 | 1.08 | 0.56 | 21.2 |
| Core 4 (8-10cm) | - | - | - | | | |
| Core 4 (10-12cm) | - | - | - | | | |
| Core 4 (12-15cm) | - | - | - | | | |
| **Avg.** | **0.53** | **-0.28** | **19.55** | **1.07** | **0.65** | **21.67** |
| **Std. Dev.** | **0.01** | **0.01** | **0.39** | **0.00** | **0.01** | **0.35** |
| | | | **19.71** | | | **21.92** |
| | | | **0.79** | | | **0.66** |



## 4. Conclusions

For this study, we demonstrate the commonly used $TEX_{86}$ paleoclimate proxy can become heavily impacted by the ocean floor archaeal community. For the Cathedral Hill vent site at Guaymas Basin, the lipids sourced from these sediments resulted in $TEX_{86}$ reconstructed temperatures that record conditions of the advecting porewaters. However, the impact appears to result from a combination of source inputs, their diagenetic and catagenetic alteration, and further overprint by the additions of lipids from the ocean floor sedimentary archaeal community that has adapted to the high-temperature conditions of the vent fluids by producing more cyclized ring moieties to rigidify their cellular membranes. Together, these processes resulted in absolute $TEX_{86}^{H}$ temperature offsets of up to 4 °C based on calibrations closely suited to the latitudinal position of Guaymas Basin. Such large offsets could be meaningful to paleoclimate reconstructions (i.e. global changes by 2–4 °C mean completed deglaciation). As such, we further present a method to correct the overprints by both water column and subsurface archaeal community's using IPLs extracted from both of these sources. Although, we have not been able to test this model with lipid inputs from the overlying water column, we have demonstrated its effectiveness at removing sediment sourced overprints, which may not be unique to hydrothermal systems. This approach should be capable of being extended to all near-surface marine sediment systems and may improve the quality of calibration models or climate reconstructions that are based on $TEX_{86}$ measures.

**Conflicts of Interest**
The authors declare no conflict of interest.

**Supplementary information**
Supplementary material related to this article can be found on-line at https://doi.org/......



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

Comprehensive glycerol ether lipid fingerprints through a novel reversed phase
liquid chromatography–mass spectrometry protocol. Organic Geochemistry, 65, 53–
965      62.