# Peer review of "The influence of near surface sediment hydrothermalism on the TEX86 tetraether lipid-based proxy and a new correction for ocean bottom lipid overprinting"

_Biogeosciences, 2021_

## Author Response (AR1)

Dear Dr. Ventura and colleagues:

Thanks for submitting your paper to Biogeosciences. The topic is of interest to a wide audience, and I have read it with pleasure. However, after your paper has been made available in the Biogeosciences discussion forum, some issues were brought to my attention (see below). These issues need to be resolved, or at the least require your attention before I will contact referees to evaluate your manuscript further.

1. Sections 2.2 and 2.3 of this BG submission are almost 100 % identical to sections 2.3 and 2.5 of the OG paper (Bentley et al, 2021) that has been published. (It appears that through an unfortunate timing the Ithenticate software did not detect this overlap at the time of submission and my preliminary evaluation). The rest of the text has no substantial overlaps.

   These two sections have been completely re-written to remove the overlap.

2. The introduction of the paper needs attention: some of the referencing is inaccurate and I recommend articulating clearer the relations with the Bentley et al. paper (in terms of differences in focus, overlap in sampling sites, methods, etc).

   The introduction has been heavily revised in content and style. Referencing has been fully reviewed and a number of new citations were added to the revised text. Several sentences have also been added to indicate what the Bentley et al. 2022 paper was focused on and how the BG paper builds on that work.

3. If you use maps that have been published before, this should be clearly communicated to the reader and permission asked if needed (because of copy-right issues). Most scientists slightly modify maps for this reason.

   The map figure has been further altered to maximize the print space of the published paper form. The source of prior work to build apon this map have been provided in the figure caption.

4. Clearly communicate to the reader the similarities and differences in samples, sites, methods, etc. between this and your prior paper on lipid biomarkers in the Cathedral Hill hydrothermal vent system.

   This has been done in the last paragraph of the introduction (section 1).

Additional changes have been made to the paper:

1. The  references section to add new citations and to remove those that no longer fit with the revised text.

2. The introduction has been amended to improve the flow of the discussion surrounding the TEX$_{86}$ proxy. All references were checked and updated to ensure they are accurately reflecting the prior work reviewed in the introduction (a through all other parts of the text).
3. Supplemental Table 1 was updated to conform to the data series used in Bentley et al., 2022.
4. Minor edits to the text were made throughout the paper to improve its readability.
5. All the paper's figures have been revised to improve their clarity.

I therefore invite you to upload a revised manuscript in which all issues identified have been solved (and others you might detect while further scrutinizing your paper). I will then consult the referees again to have a look at your revised paper.

With best regards,

Jack Middelburg, handling associate editor Biogeosciences

**Citation**: https://doi.org/10.5194/bg-2021-245-EC1

---

## Referee Report (RR1)

This study focuses on the preservation and in situ production of isoprenoid GDGTs near hydrothermal vents. It shows that GDGTs are quickly recycled but that sedimentary production of GDGTs causes an overprint, impacting proxies such as the TEX86. The authors propose a way to correct for this bias in modern systems.

I found this study presenting some interesting data which are suitable for publication. I have a few comments, however, which I like the authors to address and I detail these below. My main comment is that the concept presented is not new, i.e. the paper of Schouten et al. 2003 already showed that pelagic GDGTs are rapidly degraded in the Guaymas basin and that there is in situ production of GDGTs by thermophilic archaea causing a change in sedimentary GDGT distribution. This should be more discussed at several places in the manuscript. Further nuancing should also be that this overprint is likely unique for hydrothermal systems due to the exceptionally high degradation rates and unique lipid profile of thermophilic archaea and that the correction suggestion is likely rarely possible or needed.

Detailed comments:

L. 60 (abstract) and 663 (conclusion). I think you can use the correction in modern systems (provided all info is there) but not in paleoclimate studies due to a lack of constraints of several parameters. Furthermore, calibration studies use surface sediments (upper 1 cm), which also in your case is hardly affected by sedimentary overprint. So, which studies would actually really benefit from this correction? Can you provide an example from the literature and eg perform this correction?

L. 64-138. The introduction could be shortened in my view. The TEX86 was initially proposed as an SST proxy but ever since the 2007 publication of Huguet et al. in Paleoceanography this notion has been nuanced and plenty of publications as cited by the authors have shown that the signal is likely generated from (upper) subsurface waters (though might still be used to infer surface conditions). So lines 64 to 138 can be shortened to less than a page by just summarizing that TEX86 was initially proposed as an SST proxy but plenty of studies have shown that it is a subsurface signal. Then you can quickly move to the topic of your paper, i.e. pelagic versus sedimentary source for GDGTs.

L. 116: Sinninghe Damsté

L. 139-145. I challenge the notion that these two studies showed that in situ sedimentary production changed TEX86 values. In fact other studies (eg Lengger studies, Schouten et al., 2010) have shown that the IPLs in the study of eg Lips and Hinrichs are not derived from living Archaea but fossil IPLs because of the excellent preservation of pelagic glycosidic GDGTs (eg Xie et al., 2013, PNAS). This aspect should perhaps be more clear from the introduction, i.e. the simple presence of glycosidic GDGTs in sediments is in itself not proof of in situ sedimentary production.

L.162-181. The justification of using the Guaymas basin to determine in situ production is not only the active subsurface population but also that the GDGT fingerprint of these thermophilic sedimentary archaea is different from the pelagic background, as first established by Schouten et al. (2003) for this basin. Would be useful to clarify this here. If the distributions would have been similar, the signal would have likely been difficult to pick up.

l. 299. Outliers.

l. 302. What is 'relatively high' ? Would be good to get some quantitative information on this.

L. 315. The reported concentration is in too many significant numbers. With an error of +/- 4 ug/g sediment you should report concentrations rounded to whole numbers. Change throughout the manuscript.

L. 318-324. Would be useful to mention the estimates of degradation of GDGTs of Schouten et al. (2003) here for the same site: they actually found a difference between crenarchaeol versus other GDGTs. Indeed, what I am struck by is that the drastic change in GDGT distribution of GDGTs in Guaymas basin, in particular the dominance of GDGT4 with high temperatures, of Schouten et al. 2003 is not observed here. GDGT4 may co-elute with crenarchaeol under your UHPLC conditions so only mass spectra can distinguish the two. I presume this was done and also that if there was a co-elution that the amount of GDGT-4 was corrected for the contribution of the [M+2+H]+ ion of crenarchaeol. Can you speculate why there is this difference between the two studies?

l. 369. Schouten et al. (2010) was not the first one to use this equation to model degradation rates of lipids. Please refer to the original literature.

L. 463-467. Note that hydrous pyrolysis experiments typically last 3 days only. Pretty sure the Guaymas sediments were exposed to longer time periods and hence thermal cracking may have occurred at lower temperatures.

L. 479-489. Would be good to mention here that the same was observed in the Guaymas Basin by Schouten et al. (2003) who observed an increase in the RI of core lipid GDGTs with in situ temperature. Interestingly, they got a higher RI then observed here due mainly to a much lower relative abundances of crenarchaeol and a very high GDGT4. Can you speculate why your RI remains lower except for the deepest point in core 4?

L.527-529. I apologize for repeating myself here, but was this not already shown in Schouten et al., 2003, i.e. a replacement of pelagic GDGTs with GDGTs of thermophilic archaea? Other studies which have documented overprints of pelagic GDGTs are those of sedimentary sulfate-methane transition zones where GDGTs of AOM archaea (GDGTs 1-3) overprint GDGTs and thereby impact TEX86 values. This is one of the reasons why the MI index was developed by Zhang et al (2011) to check for this overprint. Would be good to mention this example here.

L. 626-627 and 645-647. It has been demonstrated here that in *hydrothermal systems* there is an overprint of GDGTs in sediments. This is a very important nuance as in these systems degradation rates are substantially higher due to the higher temperature (both biotic and abiotic degradation) as well as production of archaeal GDGTs with a distinctly different profile.

---

## Author Response (AR2)

Dear Jack Middleburg,

Please find attached our revised submission to BG.  This included the mark-up draft of the paper along with the final copy for our review. We have endeavoured to follow the advice of our reviewers.  Many of the comments were highly insightful and we appreciate the feedback. There were several remarks however, that we did not understand and of these we did not try to infer what the reviewer meant and left the text in its original form.  The location of these edits is indicated in the specific itemized responses provided below.

Additionally, we also took the liberty to add data to the tables that we thought were an oversight on our part during the last submission. These data include the summed totals of IPL and core lipids of GDGTs and those that were used to tally the TEX86 ratios (see mark-up of table 1). We further added a table to the Supplemental that includes the brGDGT data. Although this lipid class does not feature in our study, we do state why we do not consider the proxies based on these lipids.  Lastly, we found a computation error in our model. Equation 6 (line 576) uses the standard deviation of lipid degradation rates between GDGTs in its correction of thermogenic and heterotrophic loss impacts on TEX86.  The $S^2$ value had incorrectly been set to 0.11 (the measure of mean variance). It has since been changed to 0.20 (see Table S4).  This error, coupled with the recalculation of the model to include 2G-GDGTs (as requested by reviewer 2) has resulted in Figure 6 having slightly different $r^2$ values and a shift in the grouped reconstructed TEX86 SSTs.  No other errors in the computations were identified.  Reviewer 2 highlighted a disagreement for a time step delay in our overprint calculation. We maintain that this theoretically and computationally is more realistic (please see below for further discussion).

A final point, reviewer 2 requested a detailed discussion of archaeal community structures in the vent. This, however, was already done for Bentley et al. 2022.  The overlap would not be beneficial to this paper. We have instead mentioned that further discussion on the point is available in the sister paper. We have indicated this in the below response.

Sincerely,
Todd

Report #1
Submitted on 22 Mar 2022
Anonymous referee #1
Anonymous during peer-review:          **Yes**      No
Anonymous in acknowledgements of published article:  **Yes**      No

Recommendation to the editor
1) Scientific significance
Does the manuscript represent a substantial contribution to scientific progress within the scope of this journal (substantial new concepts, ideas, methods, or data)?
Excellent          Good    **Fair**    Poor
2) Scientific quality
Are the scientific approach and applied methods valid? Are the results discussed in an appropriate and balanced way (consideration of related work, including appropriate references)?

Excellent        Good    **Fair**    Poor
3) Presentation quality
Are the scientific results and conclusions presented in a clear, concise, and well structured way (number and quality of figures/tables, appropriate use of English language)?
Excellent        **Good**    Fair    Poor

For final publication, the manuscript should be
accepted as is
accepted subject to technical corrections
accepted subject to minor revisions
**reconsidered after major revisions**
rejected

Were a revised manuscript to be sent for another round of reviews:
**I would be willing to review the revised manuscript.**
I would not be willing to review the revised manuscript.

Suggestions for revision or reasons for rejection (will be published if the paper is accepted for final publication)
See comments in pdf
Referee Report: bg-2021-245-referee-report.pdf

This study focuses on the preservation and in situ production of isoprenoid GDGTs near hydrothermal vents. It shows that GDGTs are quickly recycled but that sedimentary production of GDGTs causes an overprint, impacting proxies such as the TEX86. The authors propose a way to correct for this bias in modern systems.

I found this study presenting some interesting data which are suitable for publication. I have a few comments, however, which I like the authors to address and I detail these below. My main comment is that the concept presented is not new, i.e. the paper of Schouten et al. 2003 already showed that pelagic GDGTs are rapidly degraded in the Guaymas basin and that there is in situ production of GDGTs by thermophilic archaea causing a change in sedimentary GDGT distribution. This should be more discussed at several places in the manuscript.

We realize the oversight and have made several modifications to the text.  For example, line 148 of the introduction now reads, "Building on the results of Schouten et al. (2003), it was observed that these lipids can become heavily turned over in the hotter portions of the vent site where they rarely survive long enough to become cracked into hydrocarbon biomarkers such as biphytanes and derivatives of biphytanes."

Further nuancing should also be that this overprint is likely unique for hydrothermal systems due to the exceptionally high degradation rates  and unique lipid profile of thermophilic archaea and that the correction suggestion is likely rarely possible or needed.

That may well be the case, but we suggest that this latter point is still conjecture.  Bentley et al (2022) presents a compelling case that archaeal lipids have a global preservation anomaly compared with other lipid classes.  Ecological evolutionary changes may account for some, but not all, of the lack of archaeal lipids in paleo sediments.

Detailed comments:

L. 60 (abstract) and 663 (conclusion). I think you can use the correction in modern systems (provided all info is there) but not in paleoclimate studies due to a lack of constraints of several parameters. Furthermore, calibration studies use surface sediments (upper 1 cm), which also in your case is hardly affected by sedimentary overprint. So, which studies would actually really benefit from this correction? Can you provide an example from the literature and eg perform this correction?

The correction has been made to limit the expected use of this model to modern systems has been made.

L. 64-138. The introduction could be shortened in my view. The TEX86 was initially proposed as an SST proxy but ever since the 2007 publication of Huguet et al. in Paleoceanography this notion has been nuanced and plenty of publications as cited by the authors have shown that the signal is likely generated from (upper) subsurface waters (though might still be used to infer surface conditions). So lines 64 to 138 can be shortened to less than a page by just summarizing that TEX86 was initially proposed as an SST proxy but plenty of studies have shown that it is a subsurface signal. Then you can quickly move to the topic of your paper, i.e. pelagic versus sedimentary source for GDGTs.

L. 116: Sinninghe Damsté  It is unclear what is being requested here.

L. 139-145. I challenge the notion that these two studies showed that in situ sedimentary production changed TEX86 values. In fact other studies (eg Lengger studies, Schouten et al., 2010) have shown that the IPLs in the study of eg Lips and Hinrichs are not derived from living Archaea but fossil IPLs because of the excellent preservation of pelagic glycosidic GDGTs (eg Xie et al., 2013, PNAS). This aspect should perhaps be more clear from the introduction, i.e. the simple presence of glycosidic GDGTs in sediments is in itself not proof of in situ sedimentary production.

Point taken and the information is now corrected.

L.162-181. The justification of using the Guaymas basin to determine in situ production is not only the active subsurface population but also that the GDGT fingerprint of these thermophilic sedimentary archaea is different from the pelagic background, as first established by Schouten et al. (2003) for this basin. Would be useful to clarify this here. If the distributions would have been similar, the signal would have likely been difficult to pick up.

 The point has been added to the introduction.

l. 299. Outliers. We do not know what this is referring to.

l. 302. What is 'relatively high' ? Would be good to get some quantitative information on this.

The rate is provided.

L. 315. The reported concentration is in too many significant numbers. With an error of +/- 4 ug/g sediment you should report concentrations rounded to whole numbers. Change throughout the manuscript.

While this may be true for many instrumental set-ups. We feel it is better to stay consistent with Bentley et al. (2022), which underwent a long and extensive peer review process.

L. 318-324. Would be useful to mention the estimates of degradation of GDGTs of Schouten et al. (2003) here for the same site: they actually found a difference between crenarchaeol versus other GDGTs. Indeed, what I am struck by is that the drastic change in GDGT distribution of GDGTs in Guaymas basin, in particular the dominance of GDGT4 with high temperatures, of Schouten et al. 2003 is not observed here. GDGT4 may co-elute with crenarchaeol under your UHPLC conditions so only mass spectra can distinguish the two. I presume this was done and also that if there was a coelution that the amount of GDGT-4 was corrected for the contribution of the [M+2+H]+ ion of crenarchaeol. Can you speculate why there is this difference between the two studies?

All detected lipids are quantified based on the two prominent adducts [M+2+H]+ and [M+2+NH4]+. Schouten et al. 2003 did not examine lipids from Cathedral Hill. As Guaymas Basin is host to many vent sites with different microbial communities and heat flow conditions it would be rather surprising that the two studies would find identical lipidomes.  Please see Bentley et al. 2022 for further information regarding archaeal lipid diversities.

l. 369. Schouten et al. (2010) was not the first one to use this equation to model degradation rates of lipids. Please refer to the original literature.

The citation is now prefaced as (e.g. ).

L. 463-467. Note that hydrous pyrolysis experiments typically last 3 days only. Pretty sure the Guaymas sediments were exposed to longer time periods and hence thermal cracking may have occurred at lower temperatures.

The point would seem to be pretty obvious.  Nonetheless we have tried to include the comment in the text.

L. 479-489. Would be good to mention here that the same was observed in the Guaymas Basin by Schouten et al. (2003) who observed an increase in the RI of core lipid GDGTs with in situ temperature. Interestingly, they got a higher RI then observed here due mainly to a much lower relative abundances of crenarchaeol and a very high GDGT4. Can you speculate why your RI remains lower except for the deepest point in core 4?

Bentley et al. (2022) talks much more about the RI with respect to crenarchaeol.

L.527-529. I apologize for repeating myself here, but was this not already shown in Schouten et al., 2003, i.e. a replacement of pelagic GDGTs with GDGTs of thermophilic archaea?

We would contend that it was inferred based on lipid distributions, which otherwise could be due to community composition.

Other studies which have documented overprints of pelagic GDGTs are those of sedimentary sulfate-methane transition zones where GDGTs of AOM archaea (GDGTs 1-3) overprint GDGTs and thereby impact TEX86
values. This is one of the reasons why the MI index was developed by Zhang et al (2011) to check for

this overprint. Would be good to mention this example here.

The references and statement to original discovery have been added to the concluding sentences of this section.

L. 626-627 and 645-647. It has been demonstrated here that in hydrothermal systems there is an overprint of GDGTs in sediments. This is a very important nuance as in these systems degradation rates are substantially higher due to the higher temperature (both biotic and abiotic degradation) as well as production of archaeal GDGTs with a distinctly different profile.

The conclusion has been re-written to hopefully convey this point.

Report #2
Submitted on 06 Apr 2022
Anonymous referee #2
Anonymous during peer-review:          **Yes**          No
Anonymous in acknowledgements of published article:  **Yes**          No

Recommendation to the editor
1) Scientific significance
Does the manuscript represent a substantial contribution to scientific progress within the scope of this journal (substantial new concepts, ideas, methods, or data)?
Excellent          Good    **Fair**    Poor
2) Scientific quality
Are the scientific approach and applied methods valid? Are the results discussed in an appropriate and balanced way (consideration of related work, including appropriate references)?
Excellent          Good    **Fair**    Poor
3) Presentation quality
Are the scientific results and conclusions presented in a clear, concise, and well structured way (number and quality of figures/tables, appropriate use of English language)?
Excellent          Good    **Fair**    Poor

For final publication, the manuscript should be
accepted as is
accepted subject to technical corrections
accepted subject to minor revisions
**reconsidered after major revisions**
rejected

Were a revised manuscript to be sent for another round of reviews:
I would be willing to review the revised manuscript.
**I would not be willing to review the revised manuscript.**

Suggestions for revision or reasons for rejection (will be published if the paper is accepted for final publication)

Manuscript Review BG-2021-245

"The influence of near surface sediment hydrothermalism on the TEX86 tetraether lipid-based proxy and a new correction for ocean bottom lipid overprinting" by Bentley et al.

In this manuscript, Bentley et al. use data recently published elsewhere to assess the potential overprint of sedimentary tetraether production on the TEX86 proxy near a hydrothermal vent system. The authors detect substantial loss of tetraethers in some of the investigated sediment cores, but find no selective degradation of specific tetraethers. Nonetheless, TEX86 does not report the expected SST and instead seems to be influenced by benthic production. Finally, the authors suggest a model that corrects for the influence of benthic production and diagenesis/catagenesis.

In my opinion, this manuscript needs major revisions before it can be published. The text suffers from a lack of specificity when defining terms or laying out arguments, which often makes very difficult to assess any implications of arguments. Furthermore, although benthic production is identified as the major control on TEX86 overprints, there is a surprising lack of discussing ecological controls on TEX86 (i.e., community composition and growth conditions/substrate availability/habitat; these factors are indeed mentioned in the introduction). This aspect appears to be quite important given that the suggested correction – while successful at removing a certain degree of overprint – fails at reconstructing the actual measured SSTs, an aspect not given any further attention.

Comments
l. 34: The sentence should be rephrased to read "…of tetraether lipids produced by archaea and bacteria in soils and sediments…" or something similar. Thaumarchaeal GDGT proxies are not readily applied to soil samples (I guess the idea here is a comparison with bacterial brGDGTs?).

The correction to the wording in the sentence has been made.

l. 38: please specify "environmental and/or geochemical variations". Done.
l. 39: do temperature changes have to be driven by climate change? Other mechanisms could affect seasonality or temperature depth gradients (expansion/compression of the thermocline). Correct. We have changed the wording of this sentence to include oceanographic effects.
l. 45: the phrase "…across TEX86 GDGT lipid classes" is confusing here since IPLs have not been introduced yet. The word classes has been deleted.
l. 48: delete "when it is"? Done.
l. 50: TEXH86 has not yet been introduced – The sentences above do discuss the ratio.
l. 59: please specify "geochemical and physical conditions".
l. 63: how do you assess that TEX86 is "the most widely used" archaeal paleotemperature proxy? The word most has been deleted.
l. 64: the reference to Tab. 1 should probably be deleted here? Done.
l. 68: given that the entire introductory paragraph places an emphasis on marine systems (SSTs, ocean bottom sediments, marine planktonic archaea…), I would suggest to delete the reference to lake sediments. Done.
l. 72: Blessing et al.? The typo has been corrected.
l. 72 and 97: "TEX86-based lipids"? The index is based on lipids, not vice versa. The change has been made.
l. 77: is there a reference that would support this statement? The sentence is linked to references and more follow in the proceeding discussion.

l. 82: change to "Mg/Ca or clumped …"? Done.

l. 84: introduce alkenone paleothermometry for readers not familiar with Uk'37. Done.

l. 92: Ho & Laepple (2016) do not discuss benthic production of GDGTs and there is substantial criticism of the 0-900m depth calibration suggested by these authors (see https://doi.org/10.1038/ngeo2997)

l. 99-101: highly controversial. That is the reason for this sentence being prefaced as "may also".

l. 107: the use of "deeper waters" may be confusing for some readers who understand the qualifier to indicate depths below the thermocline/below the ammonium maximum. The sentences has been changed to read: Here $TEX_{86}$ dissimilarities appear to be driven by increases in the relative abundances of the GDGT-2 and isomers of crenarchaeol (see Lui et al., 2018; Sinninghe Damsté et al., 2018) coupled with decreasing abundances of GDGT-1 and GDGT-3 below the thermocline and the ammonium maxima of the water column thereby producing a systematic reconstructed SST bias for deep-water surface sediments.

l. 116: this sentence should be more specific, Kim et al. do not calibrate against 0-900m depth. The citation is removed.

l. 120: delete "by" Done.

l. 122: abbreviation IPL not yet introduced in main text. Corrected.

l. 130-134: sentence could be split into two more digestible sentences. Done.

l. 139: "elevated sedimentation rates" in comparison to? Elevated is changed to high.

l. 153-154: This sentence suggests that additional measurements were conducted for this study rather than specifically stating that the data from Bentley et al. (2022) are used. The sentence has been qualified to improve clarity.

l. 213: The statement that TEXH86 is "for sediments outside the polar region" is somewhat misleading given the 15°C threshold. Changed to low latitudes.

l. 235: why would MilliQ water be used for the final 3 "washes", if the mixture is not liquid-liquid extracted again? This section has been slightly modified for improved clarity.

l. 236: "gentle stream"? The description seems OK to us.

l. 255-256: which mode was used? Targeted/untargeted? And what is the resolution of the Q-ToF? The sentence has been modified to now read: The mass spectrometer was set to a 100–3000 m/z scan range in positive mode in an untargeted method with 10 ppb resolution to simultaneously resolve both archaeal IPLs and CLs.

l. 274: please express the reproducibility in %. Given that many GDGTs or IPLs have concentrations below 1µg/g sediment (Tab. S1 and S2), the absolute deviation is not informative (there are few IPLs that occur at >4µg/g). We do not understand what is being asked here.

l. 282: heterotrophic loss is not addressed in this section. Now change to turnover.

l. 285-287: brGDGTs not mentioned above and numbering (typically Roman numerals) may be confusing here. This has been corrected and a table with the data has been added to the supplemental.

l. 287-288: in Fig. 2, the concentrations in core 3 do not decrease systematically. The working has change to nearly all.

l. 290: please add information how the age of the sediments was determined, fallout radionuclides? Age estimates were based on expected basin sedimentation rates (Bentley et al. 2022).

l. 299: specify "extreme vent fluid conditions" This has been changed to Due to the high temperature conditions of the vent fluids at Cathedral Hill.

l. 303: maybe rather "contain 0-4 cyclopentyl moieties…" We are not seeing the purpose of this change.

l. 306: please add regression/correlation coefficient (or other statistical means) to demonstrate the "tight control" of porewater temperature. Done.

l. 310-315: why are 1G-GDGTs with abundances of ~8 µg/g DW in core 4 used for the study, but 2G-GDGTs with abundances of ~7 µg/g DW are not further considered? Maybe expressing the abundances in % would be more convincing here. This has been corrected to include the additional lipid class.

l. 317-320: BIT, CBT, MBT not introduced yet. Overall, these two sentences have very limited meaning, please be more specific. Which paleoclimate records (these proxies are used for different purposes), what is "environmentally scaled loading" (which parameters), what are "other ocean floor sediment systems" (this could mean anything from lateral influx to benthic production, I guess)?

l. 323: when referring to "parallel lines", you actually mean to refer to similar slopes? This has now been made more clear.

l. 332: there seems to be something wrong with the numbering of the equations throughout the text. This has been fixed by eliminating the equation numbers from Table 1

l. 339-341: it would be helpful to quickly summarize the results here/provide some statistical analysis, so the reader can follow the argument made.

l. 340: "similar slopes" Correction made.

l. 342: RI and MI not introduced yet. Correction made.

l. 346-347: this trend is primarily driven by the TEX86 values obtained for core 1 whereas core 2 displays TEX86 values mostly in agreement with the values determined for the habitable zone. It would be nice to see this aspect and possible explanations for this pattern addressed in the text. Do these data points correspond to the highest porewater temperatures? This comment does not reflect our data. Core 2 has 3 samples that are in the habitable zone. The rest of the core samples fall outside that condition. The trend is predominantly formed by cores 1 and 2.

l. 353-354: please be more specific, which isotope systems? Carbon has been added for clarification.

l. 356: cGDGTs not introduced yet. Now corrected.

l. 358: MI has been used in other than cold seep systems as well. Somewhere in this paragraph, it should be mentioned that contributions from Euryarchaeota are implied in such high-MI settings. Noted.

l. 366: please be more specific, sulfate reduction rates? Not clear what the reviewer is looking for in this revision. AOM to our knowledge is a coupled system of methanogenesis and microbial sulfate reduction. So the metabolism is drawing on carbon and sulfur.

l. 367-369: I would not define 0.2-0.38 as "very low" values. How is any thermal control tested, please be more specific (include simple statistical analysis such as regression/correlation coefficients). The word very has been deleted. Regression statistics are presented on the figures.

l. 369: what does "this" refer to, the MI values or the lack of thermal controls? This has been replaced with low values.

l. 378: please add reference when citing the upper thermal limit of life. Done.

l. 396: McClymont et al. (2012) used POM from sediment traps for this purpose, not sediments. Correction made.

l. 408: if they were determined by vertical fluxes alone. Unclear what the requested change is.

l. 411: the "period of ~37.5 to 75 yrs" refers to which depth interval? This was provided.

l. 412: the "shallow-surface sample" is typically described as core-top. The change is made.

l. 415: please specify "larger ranges". The range is now added to the text.

l. 417: 10–21 cmbsf, i.e., the non-habitable zone mentioned above. Unclear what is being asked here.

l. 419-420: this could also be triggered by other processes than temperature that influence the archaeal community composition. The sentence is now changed to read "The fundamental driver for the proxy's is likely influenced by the archaeal community composition that is responding to their exposure to *in situ* vent fluid temperatures (Figure 4)."

l. 442-443: alternative ecological controls should be listed and discussed here (I assume data such as O2 or NH4 concentrations are not available to be compared to the TEX86 values?). Correct.

l. 451: abbreviation IPLTEX86 should be introduced. Done.

l. 454-459: the 1G-GDGT-derived TEX86 values can again also be influenced by other ecological factors than temperature alone. We consider the strong thermal gradients of the porewater to be the primary

controls on the ratio, how that intern is reflected in the subsurface archaeal community taxonomy is outside the scope of this project.

l. 458: the "living lipid pool" – it has been well established that IPLs are much more persistent than initially thought and that sedimentary IPL pools can contain substantial fossil components. At vent temperatures of +100°C in sulfidic conditions, headgroups will not be stable outside the host cell for any length of time. Therefore it is correct to call this the living lipid pool.

l. 461-480: I am missing a discussion about the other potential ecological controls on TEX86 values and, as is, the discussion only distinguishes 'living' and 'dead' archaea, but does not address the influence on TEX86 through changes in the contributions of Thaumarchaeota / Euryarchaeota (ANME?)/Bathyarchaeota etc. as well as potential ecotype contributions (even if the observed TEX86 trends are entirely determined by temperature, these archaea may not have the same response to adapt their membrane to temperature). The sentence has been reworded to read: In this regard, 1G-iGDGT IPLTEX86 ratio appears to also largely measure in situ porewater temperatures but may also be further impacted by the archaeal community ecology of the vent system.

l. 479: if TEX86 "may entirely record ocean bottom sediment porewater temperatures", why are the calculated SSTs not maxed out in all samples (e.g., at 0.9=40°C)? We did not say that. If it does, however, than TEX86 could also be used (calibrated) as a porewater thermometer, which would be immensely useful.

l. 481: it would be very helpful if the symbols in Fig. 5C would allow the reader to differentiate the origin of the samples in the clusters. One would assume that cluster C is only made up of samples from the non-habitable zone, but this does not seem to be the case (includes 4 samples from core 3, which is 'habitable' in its entirety. Are there any trends, are these the lower samples or is there no pattern whatsoever? We consider the attempt to draw out extra information within this plot to be a reach resulting in over-interpretation of our data.

l. 490: above, an argument is made that diagenesis is not selective. The data largely support non-selective turnover of lipids.

l. 498: Is d0-n an additional depth scaling parameter or a descriptor? Some initial info should be added here. This is a descriptor.

l. 505-509: what is the definition of the "deeper water column" here? Changed to upper water column. This information is essential to understand the implication here. Does "deeper water column" refer to the bathypelagic or indeed to the thermocline (as is done in the introduction). In case of the latter, this is a major pitfall of the suggested correction since the majority of IPLs/GDGTs is synthesized in the thermocline/in the ammonium maximum. In case of a bathypelagic origin, the contribution of GDGTs may indeed be negligible, but then I wonder why this point is not made earlier in this section and the term bWCTEX86 is not simply omitted from equation 7. From where in the water column most of the lipids are sourced is indeed not tractable in this study. It is instead lumped as being from the water column or from the sediments. We separate the two into different parts of the water column in equation 6 and 7 on theoretical grounds for future studies.

l. 514: in line 314, the number is ~7 µg/g DW. We do not follow this comment.

l. 514-515: I understand the mass balance argument, but the fact that 2G-GDGTs only come with 1 or 2 cyclopentyl moieties could actually bias TEX86 much more profoundly (thus, is not a good argument). This has been eliminated by incorporating the 2G-GDGTs.

l. 516-517: I find this sentence about 'increasing accumulation' confusing considering that there is significant loss of GDGTs in most cores. Increasing has been changed to progressive.

l. 517-524: I am getting lost here, there is too much information missing. What is the definition of "allochthonous"? Changed to water column input of lipids. Why sum the GDGT concentrations for the entire depth interval (as implied by the upper bound n) if the goal is to correct the TEX86 values in a distinct depth interval? Shouldn't the correction only scale the abundance of GDGTs in that particular

interval against the original input rather than summing the concentration in the total sediment column? No. The subsurface biosphere perpetually exists in all the unconsolidated sediments across time and is living dividing and dying. The deeper the sediment the more subsurface lipids it has collected.
Why would the core-top not already have lost some of the IPLs? The diagenetic time step would be lost. We admit that the offset is not perfect. It is not calibrated to the exact sedimentation rate of this system (which is not determinable by us). So the offset represents the estimated time delay for IPL hydrolysis. Sedimentation rates of 0.4–0.2 cm/yr imply that a 0-2cm core-top integrates 5-10 years of deposition.
l. 526-532: this section is at odds with the discussion regarding the kinetic model and the last two sentences directly contradict each other "…the selectivity of lipid classes being adsorbed to a protokerogen is undeterminable. More importantly, for this site it is insignificant…". The sentence has been rewritten to remove the ambiguity.
l. 536-538: there is no discussion on IPL turnover rates in section 3.1 and only a sentence linking GDGT turnover rates to hydrothermalism (citing the findings of Bentley et al. 2022). How GDGT and IPL concentrations compare downcore would be a useful addition to Fig. 2 or at least it could be expressed here using statistical means (Kruskal-Wallis). We find this to be excessive given the trend is so obvious.
l. 557: write "partial least squares" Done.
l. 564-565: what is the explanation for this 3°C offset relative to the actual SST? This seems to be an important discussion to be had given that the suggested TEX86-correction effectively accounts for a similar offset (some 3-4°C). This has now been eliminated do the reworking of the model.
l. 566: how is a "high degree of influence" assessed? It will be worthwhile to compare the offset in light of the calibration residuals. This request would entail a great deal more text to explain the statistics. Residual analysis could be useful, but it would also make for a more complicated write-up for what is not the primary points of the paper.
l. 568-569: please rephrase this sentence to be more meaningful. The fact that the "microbial community influences TEX86 measurements" is the reason it was proposed as a proxy. Be more specific. Correction made to mark benthic community.
l. 575: The caption should state that reconstructed SST are combined SS+WC SSTs at stated in the text. Done.
l. 597-598: what is the meaning of "global changes by 2–4°C mean completed deglaciation" and, more importantly, which reference does this statement refer to?
l. 599: above, it is explicitly stated that the deeper water column is not considered in this study. This is changed to upper water column.
l. 601-604: the overprint in other sedimentary systems may likely be much lower, especially in core-top sediments (less vivid benthic communities in non-seep settings, substantially less diagenesis/catagenesis) The comment is noted, but is a point of reflection for all readers of this paper.

Figures
Overall, the quality of the Excel figures is not great. The data in Figures 3-6 will not be discernible for people with deuteranopia, I would recommend either using different colors or different symbols to differentiate cores. The grid lines in Figures 3-6 also obscure the data (I don't think they are needed, but if the authors prefer them, they should recede into the background). References to background colors also do not correspond to the colors displayed (e.g., pink in Fig. 3 or blue in Fig. 4).

Some other remarks
The use of the phrase "lipid classes" throughout the text is incorrect when referring only to differences in the biphytane chains rather than headgroups. The word class has been dropped from the text.

The phrase "loading" has a statistical connotation and may be confusing. The wording has been changed.

There is an inconsistent delimiter use throughout the text when reporting numbers (and it often suggests much higher precision than can be expected).

The authors should give the text a final read to check spelling/grammar/punctuation and identify text residuals from previous manuscript versions.

---

## Author Response (AR3)

Dear Jack Middleburg,

Thank you for the opportunity to provide a revised submission of our paper to BG. The re-submission includes the mark-up draft of the paper, this cover letter and response to the reviews, the supplemental data section (unedited), along with the final copy for review. We have endeavoured to follow the advice of our reviewers. Many of the comments were highly insightful and we appreciate the feedback. The location of these edits is indicated in the specific itemized responses provided below.

Noting comments made by BGs editorial staff about the color choice to our figures, we have taken this opportunity to also revise the figures to make them clearer for those who are color impaired. This has taken a great deal of time and we struggled with issues of whether the changes would make the figures harder to read (especially if printed in blank and white). If these changes are not deemed an improvement, we would like to request specific suggestions on how to make them acceptable.

Lastly, we also took the liberty to add an acknowledgements section to the end of the paper that includes appreciation for your efforts and those of the three reviewers.

Sincerely,

Todd

Submitted on 13 Jun 2022
Anonymous referee #3

**Anonymous during peer-review:** Yes No
**Anonymous in acknowledgements of published article:** Yes No

**Recommendation to the editor**

**1) Scientific significance**
Does the manuscript represent a substantial contribution to scientific progress within the scope of this journal (substantial new concepts, ideas, methods, or data)?

Excellent Good Fair Poor

**2) Scientific quality**
Are the scientific approach and applied methods valid? Are the results discussed in an appropriate and balanced way (consideration of related work, including appropriate references)?

Excellent Good Fair Poor

**3) Presentation quality**
Are the scientific results and conclusions presented in a clear, concise, and well structured way (number and quality of figures/tables, appropriate use of English language)?

Excellent Good Fair Poor

For final publication, the manuscript should be

**accepted as is**

accepted subject to **technical corrections**

accepted subject to **minor revisions**

reconsidered after **major revisions**

**rejected**

Were a revised manuscript to be sent for another round of reviews:

I would be willing to review the revised manuscript.

I would not be willing to review the revised manuscript.

**Suggestions for revision or reasons for rejection (will be published if the paper is accepted for final publication)**

Manuscript bg-2021-245

"The influence of near surface sediment hydrothermalism on the TEX86 tetraether lipid-based proxy and a new correction for ocean bottom lipid overprinting"

by Bentley et al.

Here the authors use novel GDGT date to determinine the potential overprint of sedimentary tetraether production on the TEX86 proxy and associated water temperatures near a hydrothermal vent system. The authors propose a way to correct for this. However, upon reading the manuscript it seems to me that this impact maybe be particularly important for hydrothermal vents. It remains unclear to what extent the processes as observed in this specific 'anomalous endmember' are more widely applicable and I am not convinced that a correction is generally needed. General statements such as those at the end of the abstract ("A model to correct the overprint signal using IPLs is therefore presented that can similarly be applied to all near-surface marine sediment systems where calibration models or climate reconstructions are made based on the TEX86 measure") or conclusion section are therefore not completely justified (see specific comments below). However, overall, I feel that the manuscript is generally well written, very interesting, and after some minor revision, can be published.

Comments:

Line 95 to 133. I question the need for these sections in the introduction. Both sections are about non-thermal influences/driving forces. This is not what the manuscript is about.

The section is quite important as a foundation to section 3.4. As such we have not removed this section, but we have considerably shortened the text. Additional comments to the end are provided below.

Figure 1. The quality of this figure needs to be improved. Suggest adding proper labels to the

indivual panels and changing the background colour of panel C to make the colours of the push core transect stand out a bit better. In addition, 'Photo' in the figure caption should be 'Photos'.

This figure (and all subsequent figures) have been revised. For figure 1, all recommended changes have been made. For the following figures, we have revised the color scheme to conform to recommendations provided by BG (https://helpx.adobe.com/photoshop/using/proofing-colors.html). The color scheme for samples follows the pallet below.

[Figure]

Table 1 (and 2). The table headings lacks detail. Suggest making clear that this is data from sediment push cores that were collected along a transect at the Cathedral Hill.

The correction is made.

Line 249 and 250. Remove 'then' (2 times).

Done.

Line 334. The abbreviation BIT, CBT and MBT are not introduced the first time used. In fact, if I am correct, they are only used once throughout the whole manuscript. I suggest removing these completely and change the sentence to ' …records such as TEX86 are based….'

The mention of these other proxies has been removed.

Figure 4. It remains unclear to me why the three panels are not of equal size? It currently suggests that panel A is of greater importance.

The figure has been revised and all panels are of the same size.

Line 489. For clarity and to avoid confusion I suggest that the authors make clearly that the clusters are grouped together based on their presence in unique thermal zones immediately and not a few lines later. In addition, it would be helpful if this information is added to the figure caption of figure 5 as well.

The suggested change has been made.

Line 498-501. Based on lipid analyses is there any evidence that would justify that the hyperthemophilic Methanopyrus kandleri could indeed contribute and would fall into cluster C?

We do not have lipid or genomic data to reasonably link the hyperthermophile. So as this is speculation, we have decided to remove the sentence.

Figure 5. What does the 'R2 = 0.42' refer to in Panel A?

The value was to indicate the overall sample calculation. This has been removed from the revised figure.

Line 587-588. Please make clear that this is only tested in this very 'anomalous endmember' environment (line 136) and has not been tested across more 'common' environments.

The sentence has been amended following the recommendation.

Line 625-627. I feel that this final statement is not fully justified, given that it is only tested on one very specific environment, and too speculative. The authors do not present a clear case why this can/should be extended to all near-surface marine sediment anywhere in the manuscript. In fact, the manuscript does not need a statement such as this at the end of the conclusion section. It contributes only little (if anything). I therefore suggest removing it completely. The last line of the abstract (lines 53-55) should be adjusted accordingly.

We respectfully disagree with this point. Section 3.2 details an exact case where TEX86 when applied to other "non-hydrothermal" Guaymas Basin sediments and does not accurately produce SSTs. For McClymont et al. 2012, the application of $U_{37}^{k\prime}$ did produce better matched temperature reconstructions for more modern satellite derived MASST. Also, as evidenced in the introduction, there are many instances where the proxy does not work. Based on past reported literature and the outcomes of our study, we feel the final sentence of the conclusion does merit an allowance of speculation on our part. We also feel the readers of BG would like to entertain that though in reflecting on whether the presented model could be a benefit to their own research applications.

Submitted on 22 Jun 2022
Anonymous referee #1

**Anonymous during peer-review:** Yes No

**Anonymous in acknowledgements of published article:** Yes No

**Recommendation to the editor**

**1) Scientific significance**
Does the manuscript represent a substantial contribution to scientific progress within the scope of this journal (substantial new concepts, ideas, methods, or data)?

Excellent Good Fair Poor

**2) Scientific quality**
Are the scientific approach and applied methods valid? Are the results discussed in an appropriate and balanced way (consideration of related work, including appropriate references)?

Excellent Good Fair Poor

**3) Presentation quality**
Are the scientific results and conclusions presented in a clear, concise, and well structured way (number and quality of figures/tables, appropriate use of English language)?

Excellent Good Fair Poor

For final publication, the manuscript should be

**accepted as is**

accepted subject to **technical corrections**

accepted subject to **minor revisions**

reconsidered after **major revisions**

**rejected**

Were a revised manuscript to be sent for another round of reviews:

I would be willing to review the revised manuscript.

I would not be willing to review the revised manuscript.

**Suggestions for revision or reasons for rejection (will be published if the paper is accepted for final publication)**

I read the rebuttal and the revised manuscript. Overall, the authors did a reasonable job in rebutting the comments and revising the manuscript though a number of comments of both reviewers were not incorporated in the manuscript, either because the authors did not understand or considered e.g. 'excessive' or 'obvious'. That is a bit disappointing to read as comments were made in an honest effort to improve the manuscript.

Having said that, I think the manuscript has improved and I have a final few minor points which I recommend to consider before the manuscript is published.

Key point no. 3: I think this statement should be nuanced. The correction can only be applied in modern systems and hence will only be indirectly of use for paleostudies.

We have changed the statement to read: "A diagenetic correction model is presented to remove overprinting artifacts in the TEX$_{86}$ proxy."

Introduction. I asked this to be shortened as it is well known for quite some time that TEX86 is not an SST proxy. I see, however, no rebuttal to this comment and also no shortening of the introduction.

The introduction has been substantially shortened.

l.98-100. It has now been well established that MGII do not produce any GDGTs based on genetic studies. See Zeng et al., 2022, Nature Communications.

We have removed the sentence suggesting MGII may contribute GDGTs.

L 115. No reference given for this statement.

This sentence was removed during the revision of the introduction.

l. 307, 326. The fact that you report concentrations (or rates in line 307) in too many significant numbers in your companion paper is no excuse not to do it here. It simply is incorrect to report it in this precision. The Table 1 does a better job at this, so why not be consistent?

The correction has been made.

Line 588. Change measurements into values and add" .. in hydrothermal settings", i.e. you have demonstrated an impact of benthic microbial communities in this setting but not elsewhere (yet). I also commented previously that benthic AOM microbes also imprint the sedimentary signature and that this should be acknowledged. However, I may have missed this in the revised version.

We have changed the sentence to read: "Although, this study demonstrates the benthic microbial community can influence TEX$_{86}$ values in anomalous, end-member environments; the above model has not yet been tested across conventional ocean shelf environments."  We believe the previous mention of

AOM microbes imprinting is related to the Schouten et al 2003 study, which is referenced both in the above paragraph as well as the introduction of this paper. We also are of the understanding that AOM overprints are usually observed with archaeols, which do not factor into the calculation of the $TEX_{86}$ proxy.